# JENSEN-SHANNON DIVERGENCE BASED NOVEL LOSS FUNCTIONS FOR BAYESIAN NEURAL NETWORKS

## ABSTRACT

We aim to overcome the limitations of Kullback-Leibler (KL) divergence-based variational inference (VI) used in Bayesian Neural Networks (BNNs), which stem from the lack of boundedness of KL-divergence. These limitations include unstable optimization, poor approximation, and difficulties in approximating light-tailed posteriors, which are well documented in the literature. To overcome these limitations, we propose two novel loss functions for BNNs based on Jensen-Shannon (JS) divergences, which are more general, and one of them is bounded. We employ a constrained optimization framework to formulate these loss functions due to the intractability of the JS divergence-based VI. Further, we show that the two loss functions presented here generalize the conventional KL divergence-based loss function for BNNs. In addition to establishing stability in optimization, we perform rigorous theoretical analysis, and empirical experiments to evaluate the performance of the proposed loss functions. The empirical experiments are performed on the Cifar-10 data set with various levels of added noise and a highly biased histopathology data set. Our analysis and experiments suggest that the proposed losses perform better than the KL divergence-based loss and significantly better than their deterministic counterpart. Similar improvements by the present approach are also observed on the Cifar-100 data set. We also perform experiments on six other regression datasets and compare the performance with the existing VI approaches for BNNs.

## 1 INTRODUCTION

Despite the widespread success of deep neural networks (DNNs) and convolutional neural networks (CNNs) in numerous applications (Samarasinghe, 2016; Li et al., 2021), they suffer from *overfitting* when the data set is small, noisy, or biased (Buda et al., 2018; Thiagarajan et al., 2021). Further, due to deterministic parameters, CNNs cannot provide a robust measure of uncertainty. Without a measure of uncertainty in the predictions, erroneous predictions by these models may lead to catastrophic failures in applications that necessitate high accuracy such as autonomous driving and medical diagnosis. Several methods were developed to provide prediction intervals as a measure of uncertainty in neural networks (Kabir et al., 2018). Amongst these, Bayesian methods have gained eminence due to their rigorous mathematical foundation for uncertainty quantification through their stochastic parameters (Jospin et al., 2022; Kabir et al., 2018).

A Bayesian neural network (BNN) has stochastic parameters whose posterior distribution is learned through the Bayes rule (Tishby et al., 1989; Denker & LeCun, 1990; Goan & Fookes, 2020; Gal, 2016). Since the posterior distribution of parameters is intractable, the two most commonly used techniques to approximate them are the Variational Inference (VI) (Hinton & Van Camp, 1993; Barber & Bishop, 1998; Graves, 2011; Hernández-Lobato & Adams, 2015) and the Markov Chain Monte Carlo Methods (MCMC) (Neal, 2012; Welling & Teh, 2011). MCMC methods comprise a set of algorithms to sample from arbitrary and intractable probability distributions. Inference of posterior using MCMC algorithms can be very accurate but they are computationally demanding (Robert et al., 2018). An additional limitation of MCMC algorithms is that they do not scale well with the model size.

The VI is a technique to approximate an intractable posterior distribution by a tractable distribution called the variational distribution. The variational distribution is learned by minimizing an objective

function derived from its dissimilarity with respect to the true posterior (Blundell et al., 2015). VI methods are efficient and they scale well for larger networks and have gained significant popularity. Most of the VI techniques in the literature use the KL divergence as a measure of the aforementioned dissimilarity. However, the KL divergence is unbounded which may lead to failure during training as reported in Hensman et al. (2014); Dieng et al. (2017); Deasy et al. (2020). In addition, KL divergence is asymmetric and thus it does not qualify as a metric. Therefore, it is imperative to explore alternative divergences for VI that can alleviate these limitations.

In regards to exploring alternate divergences, Renyi's $\alpha$-divergences have been introduced for VI in Li & Turner (2016). They proposed a family of variational methods that unified various existing approaches. A $\chi$-divergence-based VI has been proposed in Dieng et al. (2017) that provides an upper bound of the model evidence. Additionally, their results have shown better estimates for the variance of the posterior. Along these lines, an f-Divergence based VI has been proposed in Wan et al. (2020) to use VI for all f-divergences that unified the Reyni divergence (Li & Turner, 2016) and $\chi$ divergence (Dieng et al., 2017) based VIs. While these recent works (Li & Turner, 2016; Dieng et al., 2017; Wan et al., 2020) mainly focused on obtaining a generalized/unified VI framework, the present work specifically attempts to alleviate the limitations (unboundedness and asymmetry) of the KL divergence-based VI through the Jensen-Shanon (JS) divergence. As a result, two novel loss functions are proposed, which outperform the KL loss in applications that require regularization. A modification to the skew-geometric Jensen-Shanson (JS) divergence has been proposed in Deasy et al. (2020) to introduce a new loss function for Variational Auto Encoders (VAEs), which has shown a better reconstruction and generation as compared to existing VAEs.

## 1.1 KEY CONTRIBUTIONS

In the present work, we propose two novel loss functions for BNNs, which are based on: 1) the skew-geometric JS divergence (denoted as JS-G) and 2) a novel modification to the generalized JS divergence (denoted as JS-A). The primary contribution of this work is that it resolves the unstable optimization issue by leveraging the boundedness of the novel JS-A divergence. We show that these JS divergence-based loss functions are generalizations of the state-of-the-art KL divergence-based ELBO loss function. In addition to addressing the stability of the optimization, through rigorous analysis we explain why these loss functions should perform better. In addition, we derive the conditions under which the proposed skew-geometric JS divergence-based loss function regularises better than that of the KL divergence. Further, we show that the loss functions presented in this work perform better for image classification problems where the data set has noise or is biased towards a particular class. In our work, we provide both closed-form and MC-based algorithms for implementing the two JS divergences. The MC implementation can include priors of any family.

The present work is different from the existing work on JS divergence-based VI Deasy et al. (2020) for the following reasons: **(i)** The JS-G divergence proposed in the previous work is unbounded like KL which is resolved by the JS-A divergence proposed in this work. **(ii)** Deasy et al. (2020) introduced the JS-G divergence-based loss for variational autoencoders (VAEs). In the present work, the distributions of parameters of BNNs are learned, which are numerous, as opposed to a small number of latent factors typically found in VAEs. **(iii)** The previous work is restricted to Gaussian priors due to the closed-form implementation, which this work overcomes through MC implementation.

## 2 MATHEMATICAL BACKGROUND

### 2.1 BACKGROUND: KL AND JS DIVERGENCES

The KL divergence between two random variables $\mathcal{P}$ and $\mathcal{Q}$ on a probability space $\Omega$ is defined as $\mathrm{KL}[p \,||\, q] = \int_{\Omega} p(x) \log \left[ p(x)/q(x) \right] dx$, where $p(x)$ and $q(x)$ are the probability distributions of $\mathcal{P}$ and $\mathcal{Q}$ respectively.

The KL divergence is widely used in literature to represent the dissimilarity between two probability distributions for applications such as VI. However, it has limitations such as the asymmetric property, i.e. $\mathrm{KL}[p \,||\, q] \neq \mathrm{KL}[q \,||\, p]$, and unboundedness, i.e. the divergence is infinite when $q(x) = 0$ and $p(x) \neq 0$. These limitations may lead to difficulty in approximating light-tailed posteriors as reported in Hensman et al. (2014).

To overcome these limitations a symmetric JS divergence can be employed which is defined as $\text{JS}[p \,||\, q] = \frac{1}{2}\text{KL}\left[p \,||\, (p+q)/2\,\right] + \frac{1}{2}\text{KL}\left[q \,||\, (p+q)/2\,\right]$. It can be further generalized as,

$$\text{JS}^{A_\alpha}[p \,||\, q] = (1-\alpha)\text{KL}\left(p \,||\, A_\alpha\right) + \alpha\text{KL}\left(q \,||\, A_\alpha\right) \tag{1}$$

where, $A_\alpha$ is the weighted arithmetic mean of $p$ and $q$ defined as $A_\alpha = (1-\alpha)p + \alpha q$. Although this JS divergence is symmetric and bounded, unlike the KL divergence its analytical expression cannot be obtained even when $p$ and $q$ are Gaussians. To overcome this difficulty a generalization of the JS divergence using the geometric mean was proposed in Nielsen (2019). By using the weighted geometric mean $G_\alpha(x, y) = x^{1-\alpha}y^\alpha$, where $\alpha \in [0, 1]$, for two real variables $x$ and $y$, they proposed the following family of skew geometric divergence

$$\text{JS}^{G_\alpha}[p \,||\, q] = (1-\alpha)\text{KL}\left(p \,||\, G_\alpha(p, q)\right) + \alpha\text{KL}\left(q \,||\, G_\alpha(p, q)\right) \tag{2}$$

The parameter $\alpha$ called a skew parameter, controls the divergence skew between $p$ and $q$. However, the skew-geometric divergence in Eq. 2 fails to capture the divergence between $p$ and $q$ and becomes zero for $\alpha = 0$ and $\alpha = 1$. To resolve this issue, Deasy et al. (2020) used the reverse form of geometric mean $G'_\alpha(x, y) = x^\alpha y^{1-\alpha}$, with $\alpha \in [0, 1]$ for JS divergence to use in variational autoencoders. Henceforth, only this reverse form is used for the geometric mean. The JS divergence with this reverse of the geometric mean is given by

$$\text{JS-G}[p \,||\, q] = (1-\alpha)\text{KL}\left(p \,||\, G'_\alpha(p, q)\right) + \alpha\text{KL}\left(q \,||\, G'_\alpha(p, q)\right) \tag{3}$$

This yields KL divergences in the limiting values of the skew parameter. Note that for $\alpha \in [0, 1]$, $\text{JS-G}(p||q)|_\alpha = \text{JS-G}(p||q)|_{1-\alpha}$ which is not symmetric. However, for $\alpha = 0.5$, the JS-G is symmetric with $\text{JS-G}(p||q)|_{\alpha=0.5} = \text{JS-G}(p||q)|_{\alpha=0.5}$. The geometric JS divergences, $\text{JS}^{G_\alpha}$ and JS-G given in Eq. 2 and Eq. 3 respectively, have analytical expressions when $p$ and $q$ are Gaussians. However, they are unbounded like the KL divergence. Whereas, the generalized JS divergence $\text{JS}^{A_\alpha}$ in Eq. 1 is both bounded and symmetric.

## 2.2 BACKGROUND: VARIATIONAL INFERENCE

Given a set of training data $\mathbb{D} = \{\mathbf{x}_i, \mathbf{y}_i\}_{i=1}^N$ and test input, $\mathbf{x} \in \mathbb{R}^p$, we learn a data-driven model (e.g a BNN) to predict the probability $P(\mathbf{y}|\mathbf{x}, \mathbb{D})$ of output $\mathbf{y} \in \Upsilon$, where $\Upsilon$ is the output space. The posterior probability distribution $(P(\mathbf{w}|\mathbb{D}))$ of the parameters $(\mathbf{w})$ of BNN, can be obtained using the Bayes' rule: $P(\mathbf{w}|\mathbb{D}) = P(\mathbb{D}|\mathbf{w})P(\mathbf{w})/P(\mathbb{D})$.

Where $P(\mathbb{D}|\mathbf{w})$ and $P(\mathbf{w})$ are the likelihood term and the prior distribution respectively. The term $P(\mathbb{D})$, called the evidence, involves marginalization over the distribution of weights: $P(\mathbb{D}) = \int_{\Omega_\mathbf{w}} P(\mathbb{D}|\mathbf{w})P(\mathbf{w})d\mathbf{w}$. Using the posterior distribution of weights, the predictive distribution of the output can be obtained by marginalizing the weights as $P(\mathbf{y}|\mathbf{x}, \mathbb{D}) = \int_{\Omega_\mathbf{w}} P(\mathbf{y}|\mathbf{x}, \mathbf{w})P(\mathbf{w}|\mathbb{D})d\mathbf{w}$.

The term $P(\mathbb{D})$ in the Bayes' rule is intractable due to marginalization over $\mathbf{w}$, which in turn makes $P(\mathbf{w}|\mathbb{D})$ intractable. To alleviate this difficulty, the posterior is approximated using variational inference.

In variational inference, the unknown intractable posterior $P(\mathbf{w}|\mathbb{D})$ is approximated by a known simpler distribution $q(\mathbf{w}|\boldsymbol{\theta})$ called the variational posterior having parameters $\boldsymbol{\theta}$. The set of parameters $\boldsymbol{\theta}$ for the model weights are learned by minimizing the divergence (e.g. KL divergence) between $P(\mathbf{w}|\mathbb{D})$ and $q(\mathbf{w}|\boldsymbol{\theta})$ as shown in Blundell et al. (2015).

$$\boldsymbol{\theta}^* = \arg\min_{\boldsymbol{\theta}} \text{KL}\left[q(\mathbf{w}|\boldsymbol{\theta}) \,||\, P(\mathbf{w}|\mathbb{D})\right] = \arg\min_{\boldsymbol{\theta}} \int q(\mathbf{w}|\boldsymbol{\theta}) \left[\log \frac{q(\mathbf{w}|\boldsymbol{\theta})}{P(\mathbf{w})P(\mathbb{D}|\mathbf{w})} + \log P(\mathbb{D})\right] d\mathbf{w} \tag{4}$$

Note that the term $\log P(\mathbb{D})$ in Eq. 4 is independent of $\boldsymbol{\theta}$ and thus can be eliminated. The resulting loss function $\mathcal{F}(\mathbb{D}, \boldsymbol{\theta})$, which is to be minimised to learn the optimal parameters $\boldsymbol{\theta}^*$ is expressed as:

$$\mathcal{F}_{KL}(\mathbb{D}, \boldsymbol{\theta}) = \text{KL}\left[q(\mathbf{w}|\boldsymbol{\theta}) \,||\, P(\mathbf{w})\right] - \mathbb{E}_{q(\mathbf{w}|\boldsymbol{\theta})}[\log P(\mathbb{D}|\mathbf{w})] \tag{5}$$

This loss is known as the variational free energy or the evidence lower bound (ELBO) (Graves, 2011; Blundell et al., 2015).

## 3 METHODS

In this section, we provide a modification to the generalized JS divergence, formulations of JS divergence-based loss functions for BNNs, and insights into the advantages of the proposed loss.

### 3.1 PROPOSED MODIFICATION TO THE GENERALIZED JS DIVERGENCE

The generalised JS divergence given in Eq. 1 fails to capture the divergence between $p$ and $q$ in the limiting cases of $\alpha$ since,

$$\left[\text{JS}^{\text{A}_\alpha}(p \,||\, q)\right]_{\alpha=0} = 0; \qquad\qquad \left[\text{JS}^{\text{A}_\alpha}(p \,||\, q)\right]_{\alpha=1} = 0 \qquad (6)$$

To overcome this limitation we propose to modify the weighted arithmetic mean as $A'_\alpha = \alpha p + (1 - \alpha)q$, $\alpha \in [0, 1]$ which modifies the generalized JS divergence as,

$$\text{JS-A}[p \,||\, q] = (1 - \alpha)\text{KL}\left(p \,||\, A'_\alpha\right) + \alpha\text{KL}\left(q \,||\, A'_\alpha\right) \qquad (7)$$

Hence, this yields KL divergences in the limiting cases of $\alpha$ as,

$$[\text{JS-A}(p \,||\, q)]_{\alpha=0} = \text{KL}(p \,||\, q) \qquad\qquad [\text{JS-A}(p \,||\, q)]_{\alpha=1} = \text{KL}(q \,||\, p) \qquad (8)$$

Eq. 7 ensures that $\text{JS}(P||q) = 0$ if and only if $P = q$, $\forall \alpha \in [0, 1]$. This is necessary since the divergence is a metric to represent statistical dissimilarity.

**Theorem 1:** Boundedness of the modified generalized JS divergence
*For any two distributions $P_1(t)$ and $P_2(t)$, $t \in \Omega$, the value of the JS-A divergence is bounded such that,*

$$\text{JS-A}(P_1(t)||P_2(t)) \leq -(1 - \alpha)\log\alpha - \alpha\log(1 - \alpha), \qquad \text{for } \alpha \in (0, 1) \qquad (9)$$

The proof of Theorem 1 is presented in App. B. Due to this boundedness property of the JS-A divergence, the ensuing loss functions overcome the instability in optimization that is encountered in the KL divergence-based loss. We provide a comparison of symmetry (at $\alpha = 0.5$) and boundedness for divergences used in this work in Table 1 and in App. A and B.

Table 1: Properties of various divergences

| Divergence | Bounded | Symmetric | Analytical expression |
|:----------:|:-------:|:---------:|:---------------------:|
| KL | × | × | ✓ |
| JS-A | ✓ | ✓ | × |
| JS-G | × | ✓ | ✓ |

### 3.2 INTRACTABILITY OF THE JS DIVERGENCE-BASED LOSS FUNCTIONS FORMULATED THROUGH THE VARIATIONAL INFERENCE APPROACH

In this subsection, we demonstrate that the JS divergence-based variational inference is intractable. If the JS-G divergence is used instead of the KL divergence in the VI setting (see Eq. 4), the optimization problem becomes,

$$\boldsymbol{\theta}^* = \arg\min_{\boldsymbol{\theta}} \text{JS-G}\left[q(\mathbf{w}|\boldsymbol{\theta}) \,||\, P(\mathbf{w}|\mathbb{D})\right] \qquad (10)$$

The loss function can then be written as,

$$\mathcal{F}_{JSG}(\mathbb{D}, \boldsymbol{\theta}) = \text{JS-G}\left[q(\mathbf{w}|\boldsymbol{\theta}) \,||\, P(\mathbf{w}|\mathbb{D})\right] = (1 - \alpha)\text{KL}\left(q \,||\, G'_\alpha(q, P)\right) + \alpha\text{KL}\left(P \,||\, G'_\alpha(q, P)\right) \qquad (11)$$

Where, $G'_\alpha(q, P) = q(\mathbf{w}|\boldsymbol{\theta})^\alpha P(\mathbf{w}|\mathbb{D})^{(1-\alpha)}$. Rewriting the first and the second term in Eq. 11 as,

$$T_1 = (1 - \alpha)^2 \int q(\mathbf{w}|\boldsymbol{\theta}) \log\left[\frac{q(\mathbf{w}|\boldsymbol{\theta})}{P(\mathbf{w}|\mathbb{D})}\right] d\mathbf{w}; \quad T_2 = \alpha^2 \int P(\mathbf{w}|\mathbb{D}) \log\left[\frac{P(\mathbf{w}|\mathbb{D})}{q(\mathbf{w}|\boldsymbol{\theta})}\right] d\mathbf{w} \qquad (12)$$

A detailed derivation of terms $T_1$ and $T_2$ is given in App. C. Term $T_1$ is equivalent to the loss function in Eq. 5 multiplied by a constant $(1 - \alpha)^2$.

The term $P(\mathbf{w}|\mathbb{D})$ in $T_2$ is intractable as explained in section 2.2. Therefore the JS-G divergence-based loss function given in Eq. 11 cannot be used to find the optimum parameter $\boldsymbol{\theta}^*$ which contrasts the KL divergence-based loss function in Eq. 5. Similarly, the JS-A divergence-based loss function obtained through VI is also intractable. We address this issue of intractability in the following subsection.

### 3.3 PROPOSED JS DIVERGENCE-BASED LOSS FUNCTIONS FORMULATED THROUGH A CONSTRAINED OPTIMIZATION APPROACH

To overcome the intractability of the variational inference, we propose to use a constrained optimization framework, following Higgins et al. (2017); Deasy et al. (2020), to derive JS divergence-based loss functions for BNNs. We also show that such a loss function is a generalization of the loss function obtained through the variational inference.

Given a set of training data $\mathbb{D}$, we are interested in learning the probability distribution $q(\mathbf{w}|\boldsymbol{\theta})$ of network parameters such that, the likelihood of observing the data given the parameters is maximized. Thus, the optimization problem can be written as

$$\max_{\boldsymbol{\theta}} \mathbb{E}_{q(\mathbf{w}|\boldsymbol{\theta})} \left[\log P(\mathbb{D}|\mathbf{w})\right] \tag{13}$$

Where $\boldsymbol{\theta}$ is a set of parameters of the probability distribution $q(\mathbf{w}|\boldsymbol{\theta})$. This optimization is constrained to make $q(\mathbf{w}|\boldsymbol{\theta})$ similar to a prior $P(\mathbf{w})$. This leads to a constrained optimization problem as given below:

$$\max_{\boldsymbol{\theta}} \mathbb{E}_{q(\mathbf{w}|\boldsymbol{\theta})} \left[\log P(\mathbb{D}|\mathbf{w})\right] \qquad \text{subject to } D(q(\mathbf{w}|\boldsymbol{\theta}) \,||\, P(\mathbf{w})) < \epsilon \tag{14}$$

where $\epsilon$ is a real number that determines the strength of the applied constraint and D is a divergence measure. Following the KKT approach, the Lagrangian function corresponding to the constrained optimization problem can be written as

$$\mathcal{L} = \mathbb{E}_{q(\mathbf{w}|\boldsymbol{\theta})} \left[\log P(\mathbb{D}|\mathbf{w})\right] - \lambda(D(q(\mathbf{w}|\boldsymbol{\theta}) \,||\, P(\mathbf{w})) - \epsilon) \tag{15}$$

Since $\epsilon$ is a constant it can be removed from the optimization. Also changing the sign of the above equations leads to the following loss function that needs to be minimized. [1].

$$\widetilde{\mathcal{F}}_D = \lambda D(q(\mathbf{w}|\boldsymbol{\theta}) \,||\, P(\mathbf{w})) - \mathbb{E}_{q(\mathbf{w}|\boldsymbol{\theta})} \left[\log P(\mathbb{D}|\mathbf{w})\right] \tag{16}$$

This loss function reproduces the ELBO loss (Blundell et al., 2015) when KL divergence is used and $\lambda$ is taken as 1.

In the following, we obtain loss functions for two JS divergences, namely, the geometric JS divergence, and the modified generalised JS divergence.

#### 3.3.1 GEOMETRIC JS DIVERGENCE

Using the modified skew-geometric JS divergence (JS-G) as the measure of divergence in Eq. 16 leads to the following loss function:

$$\widetilde{\mathcal{F}}_{JSG} = \lambda \text{ JS-G}(q(\mathbf{w}|\boldsymbol{\theta}) \,||\, P(\mathbf{w})) - \mathbb{E}_{q(\mathbf{w}|\boldsymbol{\theta})} \left[\log P(\mathbb{D}|\mathbf{w})\right] \tag{17a}$$

$$= \lambda(1-\alpha) \text{ KL}(q \,||\, G'_\alpha(q, P_w)) \ + \lambda\alpha \text{ KL}(P_w \,||\, G'_\alpha(q, P_w)) - \mathbb{E}_{q(\mathbf{w}|\boldsymbol{\theta})} \left[\log P(\mathbb{D}|\mathbf{w})\right] \tag{17b}$$

Note,

$$\text{KL}(q \,||\, G'_\alpha(q, P_w)) = \int q(\mathbf{w}|\boldsymbol{\theta}) \log \frac{q(\mathbf{w}|\boldsymbol{\theta})}{q(\mathbf{w}|\boldsymbol{\theta})^\alpha P(\mathbf{w})^{1-\alpha}} d\mathbf{w} = (1-\alpha) \int q(\mathbf{w}|\boldsymbol{\theta}) \log \frac{q(\mathbf{w}|\boldsymbol{\theta})}{P(\mathbf{w})} d\mathbf{w}$$

$$\text{KL}(P_w \,||\, G'_\alpha(q, P_w)) = \int P(\mathbf{w}) \log \frac{P(\mathbf{w})}{q(\mathbf{w}|\boldsymbol{\theta})^\alpha P(\mathbf{w})^{1-\alpha}} d\mathbf{w} = \alpha \int P(\mathbf{w}) \log \frac{P(\mathbf{w})}{q(\mathbf{w}|\boldsymbol{\theta})} d\mathbf{w}$$

Hence, the loss function can be written as,

$$\widetilde{\mathcal{F}}_{JSG} = \lambda(1-\alpha)^2 \mathbb{E}_{q(\mathbf{w}|\boldsymbol{\theta})} \left[\log \frac{q(\mathbf{w}|\boldsymbol{\theta})}{P(\mathbf{w})}\right] + \lambda\alpha^2 \mathbb{E}_{P(\mathbf{w})} \left[\log \frac{P(\mathbf{w})}{q(\mathbf{w}|\boldsymbol{\theta})}\right] - \mathbb{E}_{q(\mathbf{w}|\boldsymbol{\theta})} \left[\log P(\mathbb{D}|\mathbf{w})\right] \tag{18}$$

In Eq. 18, the first term is the *mode seeking* reverse KL divergence $\text{KL}(q(\mathbf{w}|\boldsymbol{\theta})||P(\mathbf{w}))$ and the second term is the *mean seeking* forward KL divergence $\text{KL}(P(\mathbf{w})||q(\mathbf{w}|\boldsymbol{\theta}))$. Therefore, the proposed loss function offers a weighted sum of the forward and reverse KL divergences in contrast to only the reverse KL divergence in ELBO. Whereas the likelihood part remains identical. The relative weighting between the forward and the reverse KL divergences can be controlled by the parameter $\alpha$. The proposed loss function would ensure better regularisation by imposing stricter penalization if the posterior is away from the prior distribution which will be demonstrated in detail in Sec. 3.4.1. The parameters $\lambda$ [2] and $\alpha$ can be used to control the amount of regularisation.

---

[1] The constrained optimization approach-based loss functions are marked by an overhead tilde.

[2] $\lambda$ is taken as 1 for $\widetilde{\mathcal{F}}_{JSG}$ in this work unless otherwise stated.

### 3.3.2 MODIFIED GENERALISED JS DIVERGENCE

Using the modified Generalised JS divergence (JS-A) as the measure of divergence in Eq. 16 leads to the following loss function:

$$\widetilde{\mathcal{F}}_{JSA} = \lambda \, \text{JS-A}(q(\mathbf{w}|\boldsymbol{\theta}) \,||\, P(\mathbf{w})) - \mathbb{E}_{q(\mathbf{w}|\boldsymbol{\theta})}\left[\log P(\mathbb{D}|\mathbf{w})\right]$$
$$= \lambda(1-\alpha) \, \text{KL}(q \,||\, A'_\alpha(q, P_w)) + \lambda\alpha \, \text{KL}(P_w \,||\, A'_\alpha(q, P_w)) - \mathbb{E}_{q(\mathbf{w}|\boldsymbol{\theta})}\left[\log P(\mathbb{D}|\mathbf{w})\right] \quad (19)$$

Where, $A'_\alpha(q, P_w) = \alpha q + (1-\alpha)P_w$. The above equation, Eq. 19, can be expanded as,

$$\widetilde{\mathcal{F}}_{JSA} = \lambda(1-\alpha)\mathbb{E}_{q(\mathbf{w}|\boldsymbol{\theta})}\left[\log \frac{q(\mathbf{w}|\boldsymbol{\theta})}{A'_\alpha(q, P_w)}\right] + \lambda\alpha\mathbb{E}_{P(\mathbf{w})}\left[\log \frac{P(\mathbf{w})}{A'_\alpha(q, P_w)}\right] - \mathbb{E}_{q(\mathbf{w}|\boldsymbol{\theta})}\left[\log P(\mathbb{D}|\mathbf{w})\right]$$
$$(20)$$

Note that the proposed loss functions in Eq. 18 and Eq. 20 yield the ELBO loss for $\alpha = 0$ and $\lambda = 1$. The minimization algorithms for the loss functions Eq. 18 and Eq. 20 are given in the App. D

### 3.4 INSIGHTS INTO THE PROPOSED JS DIVERGENCE-BASED LOSS FUNCTIONS

To better understand the proposed JS divergence-based loss functions, we use a contrived example to compare them against the conventional KL divergence-based loss function. In the following, we explore the regularization ability of the proposed loss functions. Further insights on Monte Carlo estimates are given in App. E

#### 3.4.1 REGULARISATION PERFORMANCE OF JS DIVERGENCES

Let two Gaussian distributions $q = \mathcal{N}(\mu_q, \sigma_q^2)$ and $P = \mathcal{N}(\mu_p, \sigma_p^2)$ represent the posterior and the prior distribution of a parameter in a BNN. The KL, JS-A, and JS-G divergences are evaluated by varying the mean and variance of the distribution $q$. This emulates the learning of the network parameter during training. Fig. 1 shows that as the posterior distribution ($q$) moves away from the prior distribution ($P$), the JS divergences increase more rapidly than the KL divergence. In the case of the JS-A divergence in Fig.1b and 1d, this is achieved by a higher value of $\lambda$. This implies that a greater penalization is offered by JS divergences than the KL divergence as the posterior deviates away from the prior. Thus, by assuming small values for the means of prior distributions we can regularize better by the proposed JS divergences. In practice, zero mean Gaussian priors are widely accepted for BNNs. For such priors, higher penalization of the loss function implies pushing the parameters' mean closer to zero while learning the complexity of the data. In doing this, we can achieve better regularization. This regularization process requires finding optimal values of $\alpha$ and $\lambda$ through hyperparameter optimization. In the following subsection, we theoretically analyze the regularization performance of the JS-G divergence.

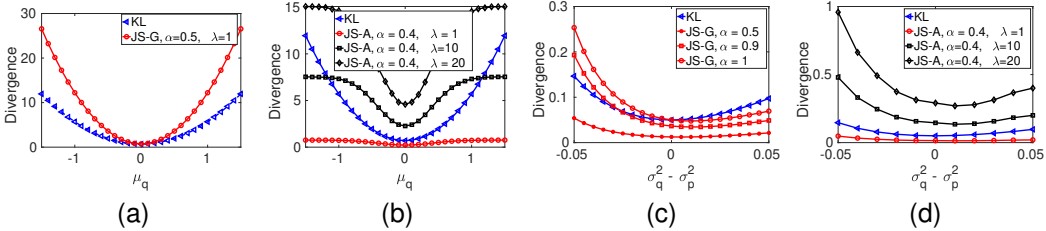

(a)          (b)          (c)          (d)

Figure 1: Comparison of the KL and the JS divergences of distributions P and q. (a) and (b) $\sigma_q^2, \mu_p, \sigma_p^2$ are fixed and $\mu_q$ is varied. (c) and (d) $\mu_q, \mu_p, \sigma_p^2$ are fixed and $\sigma_q^2$ is varied. The fixed values of the parameters are $\mu_q = 0.1, \sigma_q^2 = 0.01, \mu_p = 0, \sigma_p^2 = 0.1$

#### 3.4.2 CONDITION FOR BETTER REGULARISATION OF $\widetilde{\mathcal{F}}_{JSG}$

The above example shows that the JS-G divergence is greater than the KL for the given Gaussian distributions. To generalize it further, we propose the following theorems that hold for any two arbitrary distributions.

**Theorem 2.** *For any two arbitrary distributions $P$ and $q$ such that $P \neq q$, $\widetilde{\mathcal{F}}_{JSG} > \mathcal{F}_{KL}$ if and only if $\alpha > \frac{2\,KL(q||P)}{KL(q||P)+KL(P||q)} \in (0,\infty)$*

**Proof:** Assuming, $\widetilde{\mathcal{F}}_{JSG} - \mathcal{F}_{KL} > 0$ and from Eq. 5 and Eq. 18 we have,

$$(1-\alpha)^2 \text{KL}(q||P) + \alpha^2 \text{KL}(P||q) - \text{KL}(q||P) > 0$$
$$(\alpha^2 - 2\alpha)\text{KL}(q||P) + \alpha^2 \text{KL}(P||q) > 0$$

This leads to,

$$\alpha > \frac{2\,\text{KL}(q||P)}{\text{KL}(q||P) + \text{KL}(P||q)}$$

This proves that if $\widetilde{\mathcal{F}}_{JSG} > \mathcal{F}_{KL}$ then $\alpha > \frac{2\,\text{KL}(q||P)}{\text{KL}(q||P)+\text{KL}(P||q)}$. The converse can be proved similarly. A detailed proof is shown in the App. F

**Theorem 3.** *If $P = \mathcal{N}(\mu_p, \sigma_p^2)$ and $q = \mathcal{N}(\mu_q, \sigma_q^2)$ are Gaussian distributions and $P \neq q$, then $\frac{2\,KL(q||P)}{KL(q||P)+KL(P||q)} < 1$ if and only if $\sigma_p^2 > \sigma_q^2$.*

**Proof:** Assuming $\frac{2\,\text{KL}(q||P)}{\text{KL}(q||P)+\text{KL}(P||q)} < 1$, we get

$$\text{KL}(P||q) > \text{KL}(q||P) \tag{21}$$

Since $P = \mathcal{N}(\mu_p, \sigma_p^2)$ and $q = \mathcal{N}(\mu_q, \sigma_q^2)$, Eq. 21 can be written as,

$$\ln \frac{\sigma_q^2}{\sigma_p^2} + \frac{\sigma_p^2 + (\mu_q - \mu_p)^2}{\sigma_q^2} - 1 > \ln \frac{\sigma_p^2}{\sigma_q^2} + \frac{\sigma_q^2 + (\mu_p - \mu_q)^2}{\sigma_p^2} - 1$$

Denoting $\gamma = \frac{\sigma_p^2}{\sigma_q^2}$, we get,

$$\gamma - \frac{1}{\gamma} + \ln \frac{1}{\gamma} - \ln \gamma + \frac{(\mu_q - \mu_p)^2}{\sigma_q^2} - \frac{(\mu_p - \mu_q)^2}{\gamma \sigma_q^2} > 0$$

$$\text{or,} \quad \ln \left[ \frac{1}{\gamma^2} \exp\left(\gamma - \frac{1}{\gamma}\right) \right] + \frac{(\mu_q - \mu_p)^2}{\sigma_q^2} \left(1 - \frac{1}{\gamma}\right) > 0 \tag{22}$$

This condition Eq. 22 is satisfied only when $\gamma > 1$, which implies $\sigma_p^2 > \sigma_q^2$. Thus if $\frac{2\,\text{KL}(q||P)}{\text{KL}(q||P)+\text{KL}(P||q)} < 1$ then $\sigma_p^2 > \sigma_q^2$. This result is also observed in Fig. 1c. The converse can be proved similarly as shown in App. G.

**Corollary:** From Theorem 2 and 3: $\widetilde{\mathcal{F}}_{JSG} > \mathcal{F}_{KL}$ if $\sigma_p^2 > \sigma_q^2$ and $\forall\, \alpha \in (0,1]$ such that $\alpha > \frac{2\,\text{KL}(q||P)}{\text{KL}(q||P)+\text{KL}(P||q)}$. Where, $P$ and $q$ are Gaussians and $P \neq q$.

## 4 EXPERIMENTS

In order to demonstrate the advantages of the proposed losses in comparison to the KL loss, we performed experiments. We have implemented the divergence part of the JS-G loss and the JS-A loss via a closed-form expression and a Monte-Carlo method respectively, in these experiments.

### 4.1 DATA SETS

The following experiments were performed on two data sets: the Cifar-10 data set (Krizhevsky et al., 2009) and a histopathology data set (Janowczyk & Madabhushi, 2016; Cruz-Roa et al., 2014; Paul Mooney, 2017). To demonstrate the effectiveness of regularisation, varying levels of Gaussian noise were added to the normalized Cifar-10 data set for training, validation, and testing. We also used a histopathology data set which is highly biased towards one class. Further details on these data sets and the pre-processing steps used here are provided in App. H.

### 4.2 HYPERPARAMETER OPTIMISATION AND NETWORK ARCHITECTURE

Hyperparameters for all the networks considered here are chosen through hyperparameter optimization. A Tree-structured Parzen Estimator (TPE) algorithm (Bergstra et al., 2011) is used which is a sequential model-based optimization approach. A python library Hyperopt (Bergstra et al., 2013) is used to implement this optimization algorithm over a given search space. An optimization is performed to maximize the validation accuracy for different hyperparameter settings of the network. The results of the hyperparameter optimization are given in App. I. The architecture of all the networks used in this work follows the ResNet-18 V1 model (He et al., 2016) without the batch normalization layers. The network parameters are initialized with the weights of ResNet-18 trained on the Imagenet data set(Krizhevsky et al., 2012).

## 5 RESULTS AND DISCUSSIONS

This section presents the classification results and the performance comparison between the KL loss and the proposed JS losses. Performance evaluations on the Cifar-100 dataset along with the comparison between the proposed losses and deterministic networks, $\lambda$KL loss, and unaltered versions of JS divergences are provided in App. J. Computational costs of the losses are compared in App. K.

### 5.1 TRAINING AND VALIDATION

Three Bayesian CNNs were trained by minimizing the KL loss and the proposed JS losses. Training of the networks is done until the loss converges or the validation accuracy starts to decrease. Training

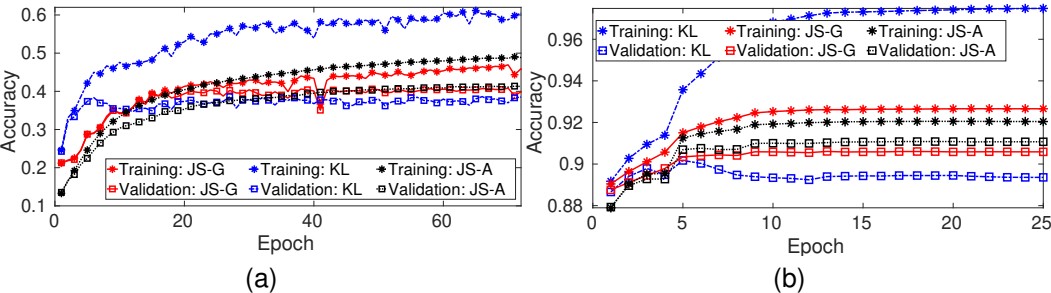

Figure 2: Training and validation of (a) Cifar-10 with added Gaussian noise (b) histopathology data set with bias.

of the Cifar-10 data set is performed with varying levels of noise intensity. Accuracy of training and validation sets for noise $\mathcal{N}(\mu = 0, \sigma = 0.9)$ is presented for both KL loss and the proposed JS losses in Fig. 2a. For the histopathology data set, a learning rate scheduler is used during training in which the learning rate is multiplied by a factor of 0.1 in the 4th, 8th, 12th, and 20th epochs. Fig. 2b shows the accuracy of training and validation of the histopathology set for the KL loss and the proposed JS losses. It is evident that the KL loss learns the training data too well and fails to generalize for the unseen validation set on both data sets. Whereas, the proposed JS losses regularise better and provide more accurate results for the validation set.

### 5.2 TESTING

Results obtained on the test sets of the Cifar-10 data set and the histopathology data set are presented in this section. The test results correspond to the epoch in which the validation accuracy was maximum. Five runs were performed with different mutually exclusive training and validation tests to compare the results of the KL loss and the proposed JS losses. The accuracy of the noisy Cifar-10 test data set at varying noise levels is presented in Fig. 3a and Fig. 3b. It is evident that the accuracy of both the proposed JS losses is better than KL for all the noise level cases. Further, the difference in accuracy between KL loss and the JS losses shows an increasing trend with increasing noise levels. This demonstrates the regularising capability of the proposed JS losses. The results of the five runs

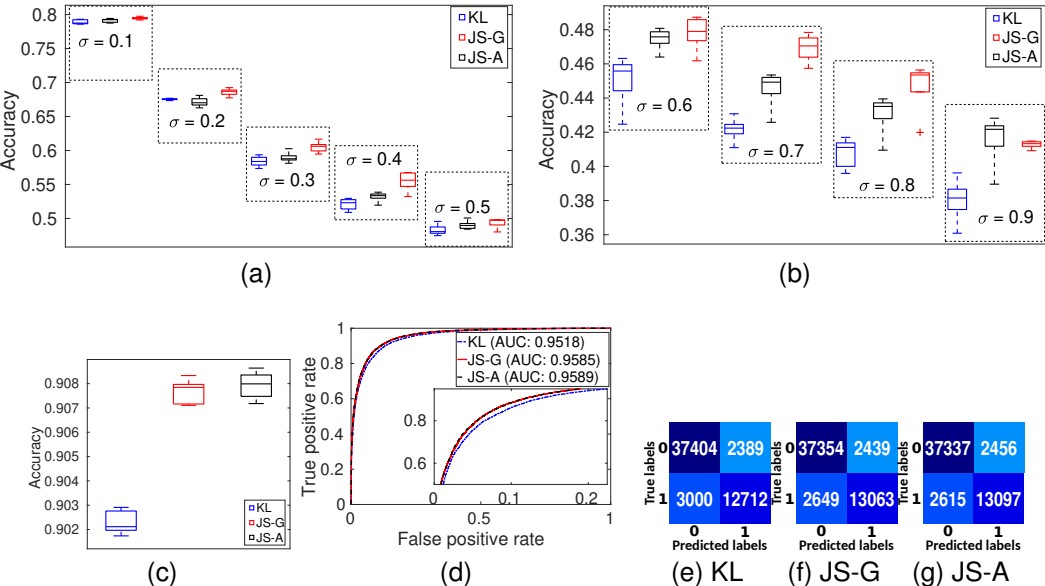

Figure 3: Accuracy on (a) and (b) the Cifar-10 test data at different noise levels (c) histopathology test data. Each box chart displays the median as the center line, the lower and upper quartiles as the box edges, and the minimum and maximum values as whiskers. (d) ROC curves and (e)-(g) Confusion matrices for different losses for the histopathology data set.

of the KL loss and the proposed JS losses on the biased histopathology data set are compared in Fig. 3c. It is evident that both the proposed JS losses perform better than the KL loss in all five runs with different training and validation sets. Since this data set is biased toward the negative class, the improvement in performance shown by the proposed JS losses is attributed to better regularisation and generalization capabilities of the loss functions. The receiver operating characteristic (ROC) curve is plotted in Fig. 3d for the classification of the histopathology data set. The proposed JS losses perform better than the KL loss in terms of the area under the curve (AUC).

The confusion matrices in Fig.3e-3g show that in addition to improving the accuracy of predictions, the proposed JS-G and the JS-A losses reduce the number of false negative predictions by 11.7% and 12.8% respectively, as compared to the KL loss. Given that the data set is biased towards the negative class, this is a significant achievement.

## 6 LIMITATIONS

The proposed loss functions have two additional hyperparameters that need to be optimized to realize their full potential, which increases computational expenses. Whenever such expenses can not be afforded, the parameters can be set to the fixed values $\alpha = 0$ and $\lambda = 1$ to recover the KL loss.

## 7 CONCLUSIONS

We summarize the main findings of this work in the following. *Firstly*, the bounded JS-A divergence introduced in this work resolves the issue of unstable optimization associated with KL divergence-based loss functions. *Secondly*, we introduced two novel loss functions for Bayesian neural networks utilizing JS divergences through a rigorous theoretical formulation. The proposed loss functions encompass the KL divergence-based loss and extend it to a wider class of symmetric and bounded divergences. *Thirdly*, better regularization performance by the proposed loss functions compared to the state-of-the-art is established analytically and numerically. *Fourthly*, empirical experiments on standard data sets having bias or with various degrees of added noise, demonstrate performance enhancement by the proposed loss functions in comparison to the existing methods.

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

## A DIVERGENCES

Two distributions, $P(x) = \mathcal{N}(0, 1)$ and $q(x) = \mathcal{N}(1, 1)$ are shown in Fig. 4 along with the divergence as a function of $x$. The area under the curve shown as a shaded region is the area to be integrated to obtain the divergence.

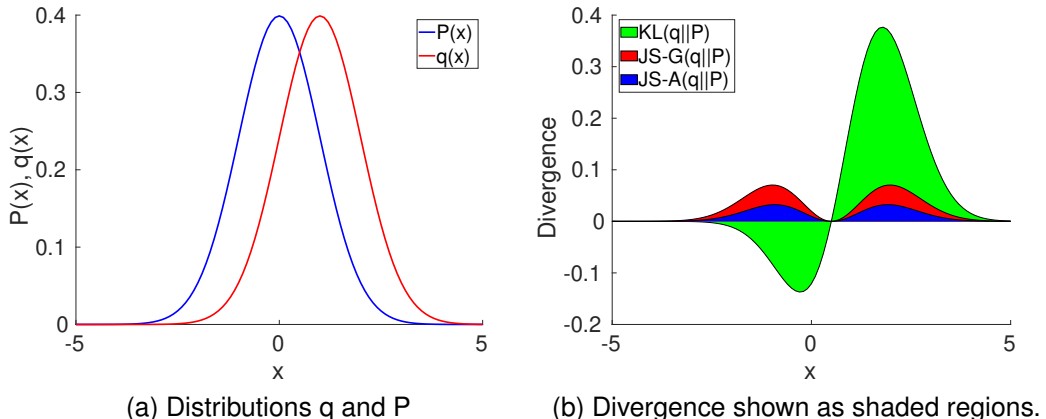

(a) Distributions q and P  (b) Divergence shown as shaded regions.

Figure 4: Depiction of KL, JS-G, and JS-A divergences for two Gaussian distributions. The area under the curves is the value of the divergences.

## B  PROOF OF THEOREM 1

**Theorem 1:** Boundedness of the modified generalized JS divergence
*For any two distributions $P_1(t)$ and $P_2(t)$, $t \in \Omega$, the value of the divergence JS-A is bounded such that,*

$$\text{JS-A}(P_1(t)||P_2(t)) \leq -(1-\alpha)\log\alpha - \alpha\log(1-\alpha), \qquad \text{for } \alpha \in (0,1)$$

**Proof:**

$$\text{JS-A}(P_1||P_2) = (1-\alpha)\int_\Omega P_1 \log\frac{P_1}{A'_\alpha}dt + \alpha\int_\Omega P_2 \log\frac{P_2}{A'_\alpha}dt$$

$$= \int_\Omega A'_\alpha\left[(1-\alpha)\frac{P_1}{A'_\alpha}\log\frac{P_1}{A'_\alpha} + \alpha\frac{P_2}{A'_\alpha}\log\frac{P_2}{A'_\alpha}\right]dt$$

$$= \int_\Omega A'_\alpha\left[\frac{(1-\alpha)}{\alpha}\frac{\alpha P_1}{A'_\alpha}\left(\log\frac{\alpha P_1}{A'_\alpha} - \log\alpha\right) + \frac{\alpha}{(1-\alpha)}\frac{(1-\alpha)P_2}{A'_\alpha}\left(\log\frac{(1-\alpha)P_2}{A'_\alpha} - \log(1-\alpha)\right)\right]dt$$

$$= \int_\Omega -(1-\alpha)P_1\log\alpha\, dt - \int_\Omega \alpha P_2\log(1-\alpha)\, dt -$$

$$\int_\Omega A'_\alpha\left[\frac{(1-\alpha)}{\alpha}H\left(\frac{\alpha P_1}{A'_\alpha}\right) + \frac{\alpha}{(1-\alpha)}H\left(\frac{(1-\alpha)P_2}{A'_\alpha}\right)\right]dt$$

$$= -(1-\alpha)\log\alpha - \alpha\log(1-\alpha) - \mathcal{H}$$

$$\leq -(1-\alpha)\log\alpha - \alpha\log(1-\alpha)$$

Where, $H(f(t)) = -f(t)\log f(t)$ and $\mathcal{H} = \int_\Omega A'_\alpha\left[\frac{(1-\alpha)}{\alpha}H\left(\frac{\alpha P_1}{A'_\alpha}\right) + \frac{\alpha}{(1-\alpha)}H\left(\frac{(1-\alpha)P_2}{A'_\alpha}\right)\right]dt$

Note, $H(f(t)) \geq 0; \quad \forall f(t) \in [0,1]$. Therefore, $\mathcal{H} \geq 0; \quad \forall t \in \Omega$

The unboundedness of the KL and the JS-G divergences is depicted through a contrived example in Fig. 5. The distribution $q$, where $q = \mathcal{N}(0, \sigma)$, is assumed to be a Gaussian with varying $\sigma$. The distribution $P$, where $P = \mathcal{U}(-5, 5)$, is assumed to be a uniform distribution with support (-5,5). The distributions $q$ and $p$ are shown in Fig. 5a. The KL, the JS-A, and the JS-G divergence of the two distributions $q$ and $P$ are evaluated using 50,000 Monte Carlo samples each. The value of the KL and the JS-G divergence explodes to infinity when the distribution $p$ is zero for a non-zero $q$ due to the effect of $\log(q/P)$. In contrast, the JS-G divergence is always bounded. This can be seen in Fig. 5b.

Numerical examples depicting Theorem-1 are shown in Fig. 6. Two distribution $q = \mathcal{N}(\mu, 1)$ and $P = \mathcal{N}(0, 1)$ are considered. The JS-A divergence is evaluated for varying $\mu$. The value of the

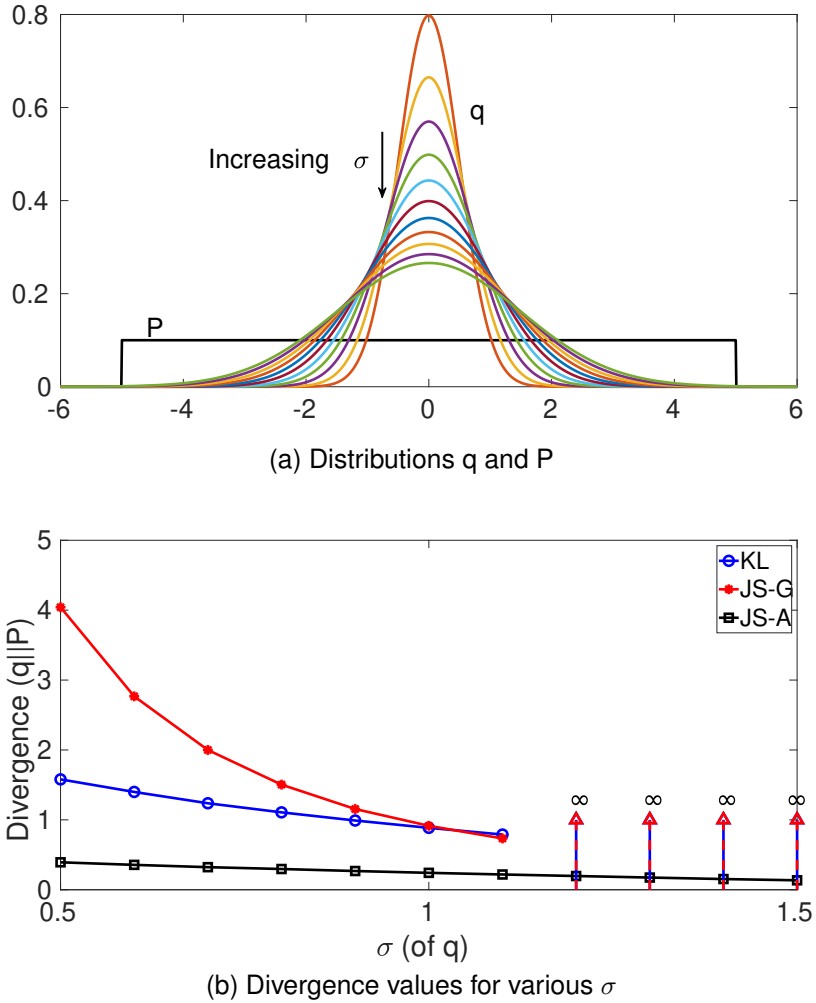

(a) Distributions q and P

(b) Divergence values for various $\sigma$

Figure 5: Depiction of the unboundedness of the KL and JS-G divergence and the boundedness of the JS-A divergence.

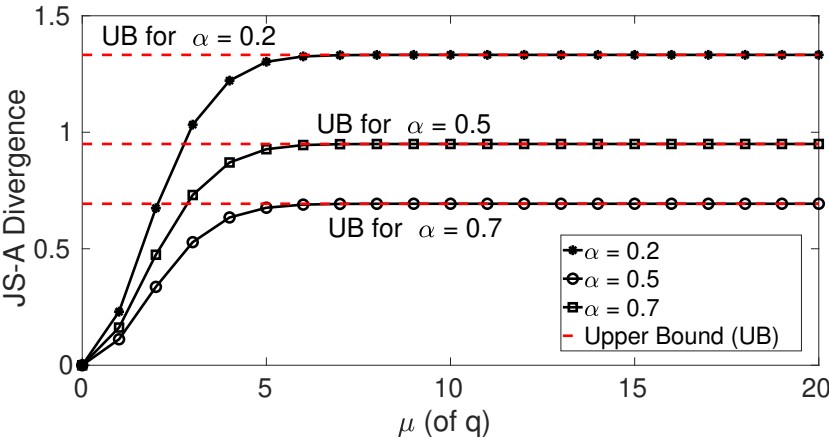

Figure 6: Upper bound of the JS-A divergence for various values of $\alpha$.

JS-A divergence increases with increasing $\mu$ until the upper bound is reached and it remains constant henceforth as seen in Fig. 6.

## C  INTRACTABILITY OF JS-BASED VI

The JS-G loss is given as

$$
\begin{aligned}
\mathcal{F}_{JSG}(\mathbb{D}, \boldsymbol{\theta}) &= \text{JS-G}\left[q(\mathbf{w}|\boldsymbol{\theta}) \,||\, P(\mathbf{w}|\mathbb{D})\right] \\
&= (1-\alpha)\text{KL}\left(q \,||\, G'_\alpha(q,P)\right) + \alpha\text{KL}\left(P \,||\, G'_\alpha(q,P)\right)
\end{aligned}
\tag{23}
$$

Where, $G'_\alpha(q,P) = q(\mathbf{w}|\boldsymbol{\theta})^\alpha P(\mathbf{w}|\mathbb{D})^{(1-\alpha)}$.
Rewriting the first term in Eq. 23 as,

$$
\begin{aligned}
T_1 &= (1-\alpha)\text{KL}\left(q \,||\, G'_\alpha(q,P)\right) \\
&= (1-\alpha)KL[q(\mathbf{w}|\boldsymbol{\theta})||q(\mathbf{w}|\boldsymbol{\theta})^\alpha P(\mathbf{w}|\mathbb{D})^{(1-\alpha)}] \\
&= (1-\alpha)\int q(\mathbf{w}|\boldsymbol{\theta})\log\left[\frac{q(\mathbf{w}|\boldsymbol{\theta})}{q(\mathbf{w}|\boldsymbol{\theta})^\alpha P(\mathbf{w}|\mathbb{D})^{1-\alpha}}\right] d\mathbf{w} \\
&= (1-\alpha)^2\int q(\mathbf{w}|\boldsymbol{\theta})\log\left[\frac{q(\mathbf{w}|\boldsymbol{\theta})}{P(\mathbf{w}|\mathbb{D})}\right] d\mathbf{w}
\end{aligned}
\tag{24}
$$

Similarly rewriting the second term in Eq. 23 as,

$$
\begin{aligned}
T_2 &= \alpha\text{KL}\left(P \,||\, G'_\alpha(q,P)\right) \\
&= \alpha\text{KL}[P(\mathbf{w}|\mathbb{D})||q(\mathbf{w}|\boldsymbol{\theta})^\alpha P(\mathbf{w}|\mathbb{D})^{(1-\alpha)}] \\
&= \alpha\int P(\mathbf{w}|\mathbb{D})\log\left[\frac{P(\mathbf{w}|\mathbb{D})}{q(\mathbf{w}|\boldsymbol{\theta})^\alpha P(\mathbf{w}|\mathbb{D})^{(1-\alpha)}}\right] d\mathbf{w} \\
&= \alpha^2\int P(\mathbf{w}|\mathbb{D})\log\left[\frac{P(\mathbf{w}|\mathbb{D})}{q(\mathbf{w}|\boldsymbol{\theta})}\right] d\mathbf{w}
\end{aligned}
\tag{25}
$$

The term $T_2$ is intractable due to the posterior distribution $P(\mathbf{w}|\mathbb{D})$.

## D  MINIMISATION OF THE PROPOSED LOSS FUNCTIONS

The proposed loss functions are of the form:

$$
\widetilde{\mathcal{F}}_{JSG} = \lambda \,\text{JS-G}(q(\mathbf{w}|\boldsymbol{\theta}) \,||\, P(\mathbf{w})) - \mathbb{E}_{q(\mathbf{w}|\boldsymbol{\theta})}\left[\log P(\mathbb{D}|\mathbf{w})\right]
\tag{26}
$$

$$
= \lambda(1-\alpha)^2\mathbb{E}_{q(\mathbf{w}|\boldsymbol{\theta})}\left[\log\frac{q(\mathbf{w}|\boldsymbol{\theta})}{P(\mathbf{w})}\right] + \lambda\alpha^2\mathbb{E}_{P(\mathbf{w})}\left[\log\frac{P(\mathbf{w})}{q(\mathbf{w}|\boldsymbol{\theta})}\right] - \mathbb{E}_{q(\mathbf{w}|\boldsymbol{\theta})}\left[\log P(\mathbb{D}|\mathbf{w})\right]
\tag{27}
$$

and,

$$\widetilde{\mathcal{F}}_{JSA} = \lambda \, \text{JS-A}(q(\mathbf{w}|\boldsymbol{\theta}) \, || \, P(\mathbf{w})) - \mathbb{E}_{q(\mathbf{w}|\boldsymbol{\theta})} \left[ \log P(\mathbb{D}|\mathbf{w}) \right] \tag{28}$$

$$= \lambda(1-\alpha)\mathbb{E}_{q(\mathbf{w}|\boldsymbol{\theta})} \left[ \log \frac{q(\mathbf{w}|\boldsymbol{\theta})}{A'_\alpha(q, P_w)} \right] + \lambda\alpha\mathbb{E}_{P(\mathbf{w})} \left[ \log \frac{P(\mathbf{w})}{A'_\alpha(q, P_w)} \right] - \mathbb{E}_{q(\mathbf{w}|\boldsymbol{\theta})} \left[ \log P(\mathbb{D}|\mathbf{w}) \right] \tag{29}$$

Where, $A'_\alpha(q, P_w) = \alpha q + (1-\alpha) P_w$.

Note that when training using mini-batches, the divergence part of the loss function is normalized by the size of the minibatch ($M$). Therefore, loss for the minibatch $i = 1, 2, 3, ..., M$ can be written as,

$$\widetilde{\mathcal{F}}_i = \frac{\lambda}{M} \, \text{D}(q(\mathbf{w}|\boldsymbol{\theta}) \, || \, P(\mathbf{w})) - \mathbb{E}_{q(\mathbf{w}|\boldsymbol{\theta})} \left[ \log P(\mathbb{D}_i|\mathbf{w}) \right] \tag{30}$$

### D.1 EVALUATION OF THE JS-G DIVERGENCE IN A CLOSED-FORM

---

**Algorithm 1** Minimization of the JS-G loss function: Closed-form evaluation of the divergence

---

**Initialize** $\boldsymbol{\mu}, \boldsymbol{\rho}$
**Evaluate** JS-G term of Eq. 26 analytically using Eq. 31
**Evaluate** $\mathbb{E}_{q(\mathbf{w}|\boldsymbol{\theta})} \left[ \log P(\mathbb{D}|\mathbf{w}) \right]$ term of Eq. 26
    Sample $\boldsymbol{\varepsilon}_i \sim \mathcal{N}(0, 1); i = 1, ..., \text{No. of samples}$
    $\mathbf{w}_i \leftarrow \boldsymbol{\mu} + \log(1 + \exp(\boldsymbol{\rho})) \circ \boldsymbol{\varepsilon}_i.$
    $f_1 \leftarrow \sum_i \log P(\mathbb{D}|\mathbf{w}_i)$
**Loss:**

    $F \leftarrow \lambda \, \text{JS-G} - f_1$

**Gradients:**

    $$\frac{\partial F}{\partial \boldsymbol{\mu}} \leftarrow \sum_i \frac{\partial F}{\partial \mathbf{w}_i} + \frac{\partial F}{\partial \boldsymbol{\mu}}$$

    $$\frac{\partial F}{\partial \boldsymbol{\rho}} \leftarrow \sum_i \frac{\partial F}{\partial \mathbf{w}_i} \frac{\boldsymbol{\varepsilon}_i}{1 + \exp(-\boldsymbol{\rho})} + \frac{\partial F}{\partial \boldsymbol{\rho}}$$

**Update:**

    $$\boldsymbol{\mu} \leftarrow \boldsymbol{\mu} - \beta \frac{\partial F}{\partial \boldsymbol{\mu}}; \qquad \boldsymbol{\rho} \leftarrow \boldsymbol{\rho} - \beta \frac{\partial F}{\partial \boldsymbol{\rho}}$$

---

In this subsection, we describe the minimization of the JS-G divergence-based loss function by evaluating the divergence in closed form for Gaussian priors. Assuming the prior and the likelihood are Gaussians, the posterior will also be a Gaussian. Let the prior and posterior be diagonal multivariate Gaussian distributions denoted by $P_N(\mathbf{w}|\boldsymbol{\theta}) = \mathcal{N}(\boldsymbol{\mu_2}, \boldsymbol{\Sigma_2^2})$ and $q_N(\mathbf{w}) = \mathcal{N}(\boldsymbol{\mu_1}, \boldsymbol{\Sigma_1^2})$ respectively [3]. The closed-form expression of the JS-G divergence between $q_N(\mathbf{w})$ and $P_N(\mathbf{w}|\boldsymbol{\theta})$ can be written as,

$$
\text{JS-G}(q_N || P_N) = \frac{1}{2} \sum_{i=1}^n \left[ \frac{(1-\alpha)\sigma_{1i}^2 + \alpha\sigma_{2i}^2}{\sigma_{\alpha i}^2} + \log \frac{(\sigma'_{\alpha i})^2}{\sigma_{1i}^{2(1-\alpha)}\sigma_{2i}^{2\alpha}} \right.
$$
$$
\left. + (1-\alpha)\frac{(\mu'_{\alpha i} - \mu_{1i})^2}{(\sigma'_{\alpha i})^2} + \frac{\alpha(\mu'_{\alpha i} - \mu_{2i})^2}{(\sigma'_{\alpha i})^2} - 1 \right] \tag{31}
$$

---

[3] Where the subscript $()_N$ indicates Gaussian distribution. $\boldsymbol{\mu}_1$ and $\boldsymbol{\mu}_2$ are n-dimensional vectors and $\boldsymbol{\Sigma_1^2}$, $\boldsymbol{\Sigma_2^2}$ are assumed to be diagonal matrices such that $\boldsymbol{\mu}_1 = [\mu_{11}, \mu_{12}, ...\mu_{1n}]^T$ and $\boldsymbol{\Sigma_1^2} = \text{diag}(\sigma_{11}^2, \sigma_{12}^2, ...\sigma_{1n}^2)$ (similarly for $\boldsymbol{\mu}_2$ and $\boldsymbol{\Sigma_2^2}$).

where,
$$(\sigma'_{\alpha i})^2 = \frac{\sigma_{1i}^2 \sigma_{2i}^2}{(1-\alpha)\sigma_{1i}^2 + \alpha\sigma_{2i}^2}; \quad \mu'_{\alpha i} = (\sigma'_{\alpha i})^2 \left[\frac{\alpha\mu_{1i}}{\sigma_{1i}^2} + \frac{(1-\alpha)\mu_{2i}}{\sigma_{2i}^2}\right]$$

Therefore, the divergence term of the proposed loss function, the first term in Eq. 26, can be evaluated by this closed-form expression given in Eq. 31. The expectation of the log-likelihood, the second term in Eq. 26, can be approximated by a Monte-Carlo sampling [4]. The details of the minimization process are given in Algorithm 1. Note, for sampling $w_i$ the reparametrization trick is used to separate the deterministic and the stochastic variables.

## D.2 EVALUATION OF DIVERGENCES VIA A MONTE CARLO SAMPLING

---

**Algorithm 2** Minimization of the JS-G and JS-A loss functions: Monte Carlo approximation of the divergence

---

**Initialize** $\boldsymbol{\mu}, \boldsymbol{\rho}$
**Approximate** $\mathbb{E}_{q(\mathbf{w}|\boldsymbol{\theta})}$ terms of Eq. 27 or Eq. 29
    Sample $\boldsymbol{\varepsilon}_i^q \sim \mathcal{N}(0,1); i = 1, ..., $ No. of samples
    $\mathbf{w}_i^q \leftarrow \boldsymbol{\mu} + \log(1 + \exp(\boldsymbol{\rho})) \circ \boldsymbol{\varepsilon}_i^q.$

    Evaluate first and third terms of Eq. 27:
    $f_1 \leftarrow \sum_i c_1 \log q(\mathbf{w}_i^q | \boldsymbol{\theta}) - c_1 \log P(\mathbf{w}_i^q) - \log P(\mathbb{D} | \mathbf{w}_i^q)$
    where, $c_1 = \lambda(1-\alpha)^2$
                  (or)
    Evaluate first and third terms of Eq. 29:
    $f_1 \leftarrow \sum_i c_1 \log q(\mathbf{w}_i^q | \boldsymbol{\theta}) - c_1 \log A_\alpha(\mathbf{w}_i^q) - \log P(\mathbb{D} | \mathbf{w}_i^q)$
    where, $c_1 = \lambda(1-\alpha)$

**Approximate** $\mathbb{E}_{P(\mathbf{w})}$ terms of Eq. 27 or Eq. 29
    Sample $\mathbf{w}_j^p \sim P(\mathbf{w}); j = 1, ..., $ No. of samples

    Evaluate second term of Eq. 27:
    $f_2 \leftarrow \sum_j c_2 \log P(\mathbf{w}_j^p) - c_2 \log q(\mathbf{w}_j^p | \boldsymbol{\theta})$
    where, $c_2 = \lambda\alpha^2$
                  (or)
    Evaluate second term of Eq. 29:
    $f_2 \leftarrow \sum_j c_2 \log A'_\alpha(\mathbf{w}_j^p) - c_2 \log q(\mathbf{w}_j^p | \boldsymbol{\theta})$
    where, $c_2 = \lambda\alpha^2$
**Loss:**

    $F \leftarrow f_1 + f_2$

**Gradients:**

$$\frac{\partial F}{\partial \boldsymbol{\mu}} \leftarrow \sum_i \frac{\partial F}{\partial \mathbf{w}_i^q} + \frac{\partial F}{\partial \boldsymbol{\mu}}$$

$$\frac{\partial F}{\partial \boldsymbol{\rho}} \leftarrow \sum_i \frac{\partial F}{\partial \mathbf{w}_i^q} \frac{\varepsilon_i}{1 + \exp(-\boldsymbol{\rho})} + \frac{\partial F}{\partial \boldsymbol{\rho}}$$

**Update:**

$$\boldsymbol{\mu} \leftarrow \boldsymbol{\mu} - \beta\frac{\partial F}{\partial \boldsymbol{\mu}}; \qquad \boldsymbol{\rho} \leftarrow \boldsymbol{\rho} - \beta\frac{\partial F}{\partial \boldsymbol{\rho}}$$

---

[4]The approximations to the loss functions are denoted by F

In this subsection, we describe the minimization of the JS divergence-based loss functions by evaluating the divergences using the Monte Carlo sampling technique. The algorithm provided in this subsection is more general as it is applicable to both the JS-G and the JS-A divergences with no restrictions on the priors. The loss functions given in Eq. 27 and Eq. 29 can be approximated using Monte Carlo samples from the corresponding distributions as shown in Algorithm 2.

## E    INSIGTHS INTO MONTE CARLO ESTIMATES

A closed-form solution does not exist for KL and JS divergences for most distributions. In cases where such a closed-form for the divergence is not available for a given distribution, we resort to Monte Carlo (MC) methods. However, the estimation of the loss function using MC methods is computationally more expensive than the closed-form evaluation as shown in 7. In addition, for networks with a large number of parameters, the memory requirement increases significantly with the number of MC samples. Therefore, utilizing the closed-form solution when available can save huge computational efforts and memory.

To estimate the number of MC samples required to achieve a similar level of accuracy of the closed-form expression, JS-G divergence of two Gaussian distributions $\mathcal{N}(5, 1)$ and $\mathcal{N}(0, 1)$ are evaluated and compared with its closed form counterpart. Fig. 7 shows the results of the comparison. It is seen that at least 600 samples are required to estimate the JS-G divergence within 5% error. This implies evaluating the loss function 600 times for a given input and back-propagating the error which requires huge computational efforts.

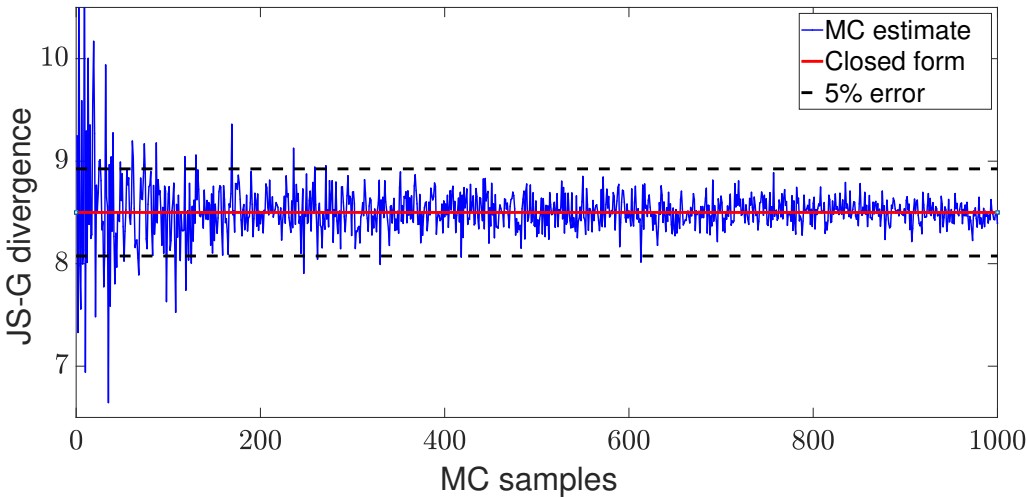

Figure 7: Comparison of MC estimates and the closed form solution of JS-G divergence demonstrating the benefit of closed form solution.

## F    DETAILED PROOF OF THEOREM 2

**Theorem 2.**  *For any two arbitrary distributions $P$ and $q$ such that $P \neq q$, $\widetilde{\mathcal{F}}_{JSG} > \mathcal{F}_{KL}$ if and only if $\alpha > \frac{2\,KL(q||P)}{KL(q||P)+KL(P||q)} \in (0, \infty)$*

**Proof:**
Assuming,

$$\widetilde{\mathcal{F}}_{JSG} - \mathcal{F}_{KL} > 0$$

From Eq.(5) and Eq.(18) from the main text we have,

$$(1 - \alpha)^2 \text{KL}(q||P) + \alpha^2 \text{KL}(P||q) - \text{KL}(q||P) > 0$$
$$(\alpha^2 - 2\alpha)\text{KL}(q||P) + \alpha^2 \text{KL}(P||q) > 0$$

This leads to,

$$\alpha > \frac{2 \text{ KL}(q||P)}{\text{KL}(q||P) + \text{KL}(P||q)}$$

This proves that if $\widetilde{\mathcal{F}}_{JSG} > \mathcal{F}_{KL}$ then $\alpha > \frac{2 \text{ KL}(q||P)}{\text{KL}(q||P)+\text{KL}(P||q)}$ .

Conversely, assuming $\alpha > r$, where $r = \frac{2 \text{ KL}(q||P)}{\text{KL}(q||P)+\text{KL}(P||q)}$, we get

$$\alpha\left[\text{KL}(q||P) + \text{KL}(P||q)\right] - 2\text{ KL}(q||P) > r\left[\text{KL}(q||P) + \text{KL}(P||q)\right] - 2\text{ KL}(q||P) \qquad (\text{since } \alpha > r)$$
$$\alpha\left[\text{KL}(q||P) + \text{KL}(P||q)\right] - 2\text{ KL}(q||P) > 0 \qquad (\text{substituting r on the right hand side})$$
$$\alpha^2\left[\text{KL}(q||P) + \text{KL}(P||q)\right] - 2\alpha\text{ KL}(q||P) > 0 \qquad (\text{since } \alpha > 0)$$
$$(\alpha^2 - 2\alpha)\text{KL}(q||P) + \alpha^2 \text{KL}(P||q) > 0$$
$$(1 - \alpha)^2 \text{KL}(q||P) + \alpha^2 \text{KL}(P||q) - \text{KL}(q||P) > 0$$
$$\widetilde{\mathcal{F}}_{JSG} - \mathcal{F}_{KL} > 0$$

This proves that if $\alpha > \frac{2 \text{ KL}(q||P)}{\text{KL}(q||P)+\text{KL}(P||q)}$ then $\widetilde{\mathcal{F}}_{JSG} > \mathcal{F}_{KL}$. Hence the theorem is proved.

Note that, since KL divergence is always non-negative for any distributions and $P \neq q$, we have $\frac{2 \text{ KL}(q||P)}{\text{KL}(q||P)+\text{KL}(P||q)} > 0$

## G    DETAILED PROOF OF THEOREM 3

**Theorem 3.** *If $P = \mathcal{N}(\mu_p, \sigma_p^2)$ and $q = \mathcal{N}(\mu_q, \sigma_q^2)$ are Gaussian distributions and $P \neq q$, then $\frac{2 \text{ KL}(q||P)}{KL(q||P)+KL(P||q)} < 1$ if and only if $\sigma_p^2 > \sigma_q^2$ .*

**Proof:**
Assuming,

$$\frac{2 \text{ KL}(q||P)}{\text{KL}(q||P) + \text{KL}(P||q)} < 1$$

We have,

$$\text{KL}(P||q) > \text{KL}(q||P) \tag{33}$$

Since $P$ and $q$ be Gaussian distributions with $P = \mathcal{N}(\mu_p, \sigma_p^2)$ and $q = \mathcal{N}(\mu_q, \sigma_q^2)$, Eq. 33 can be written as,

$$\ln \frac{\sigma_q^2}{\sigma_p^2} + \frac{\sigma_p^2 + (\mu_q - \mu_p)^2}{\sigma_q^2} - 1 > \ln \frac{\sigma_p^2}{\sigma_q^2} + \frac{\sigma_q^2 + (\mu_p - \mu_q)^2}{\sigma_p^2} - 1$$

$$\frac{\sigma_p^2}{\sigma_q^2} + \ln \frac{\sigma_q^2}{\sigma_p^2} + \frac{(\mu_q - \mu_p)^2}{\sigma_q^2} - \frac{\sigma_q^2}{\sigma_p^2} - \ln \frac{\sigma_p^2}{\sigma_q^2} - \frac{(\mu_p - \mu_q)^2}{\sigma_p^2} > 0$$

Denoting $\gamma = \frac{\sigma_p^2}{\sigma_q^2}$, we get,

$$\gamma - \frac{1}{\gamma} + \ln \frac{1}{\gamma} - \ln \gamma + \frac{(\mu_q - \mu_p)^2}{\sigma_q^2} - \frac{(\mu_p - \mu_q)^2}{\gamma \sigma_q^2} > 0$$

$$\gamma - \frac{1}{\gamma} + \ln \frac{1}{\gamma^2} + \frac{(\mu_q - \mu_p)^2}{\sigma_q^2}\left(1 - \frac{1}{\gamma}\right) > 0$$

$$\ln\left[\exp\left(\gamma - \frac{1}{\gamma}\right)\right] + \ln \frac{1}{\gamma^2} + \frac{(\mu_q - \mu_p)^2}{\sigma_q^2}\left(1 - \frac{1}{\gamma}\right) > 0$$

or,

$$\ln\left[\frac{1}{\gamma^2}\exp\left(\gamma-\frac{1}{\gamma}\right)\right]+\frac{(\mu_q-\mu_p)^2}{\sigma_q^2}\left(1-\frac{1}{\gamma}\right)>0 \tag{34}$$

The second term of Eq. 34 is greater than 0 only when $\gamma>1$. Consider the first term,

$$\ln\left[\frac{1}{\gamma^2}\exp\left(\gamma-\frac{1}{\gamma}\right)\right]>0$$

or,

$$\frac{1}{\gamma^2}\exp\left(\gamma-\frac{1}{\gamma}\right)>1$$

$\frac{1}{\gamma^2}\exp\left(\gamma-\frac{1}{\gamma}\right)=1$ for $\gamma=1$ and it is a monotonically increasing function for $\gamma>1$. This can be seen from its positive slope

$$\begin{aligned}
\frac{d}{d\gamma}\left[\frac{1}{\gamma^2}\exp\left(\gamma-\frac{1}{\gamma}\right)\right] &= \frac{1}{\gamma^2}\exp\left(\gamma-\frac{1}{\gamma}\right)\times\left(1+\frac{1}{\gamma^2}\right)-\frac{2}{\gamma^3}\exp\left(\gamma-\frac{1}{\gamma}\right)\\
&= \frac{1}{\gamma^4}\exp\left(\gamma-\frac{1}{\gamma}\right)\left[\gamma^2+1-2\gamma\right]\\
&= \frac{1}{\gamma^4}\exp\left(\gamma-\frac{1}{\gamma}\right)(\gamma-1)^2\\
&> 0 \qquad\qquad \text{for } \gamma\neq 1
\end{aligned}$$

Therefore, the first term of Eq. 34 is greater than 0 only when $\gamma>1$. Thus, the condition in Eq. 34 is satisfied only when $\gamma>1$, which implies

$$\sigma_p^2>\sigma_q^2 \tag{35}$$

Thus if $\frac{2\,\text{KL}(q||P)}{\text{KL}(q||P)+\text{KL}(P||q)}<1$ then $\sigma_p^2>\sigma_q^2$.

Conversely, assuming $\sigma_p^2>\sigma_q^2$, i.e. $\gamma>1$ consider,

$$\ln\left[\frac{1}{\gamma^2}\exp\left(\gamma-\frac{1}{\gamma}\right)\right]+\frac{(\mu_q-\mu_p)^2}{\sigma_q^2}\left(1-\frac{1}{\gamma}\right)>0$$

Which leads to,

$$\text{KL}(P||q)>\text{KL}(q||P)\qquad\text{(following the steps as above)}$$

or,

$$\text{KL}(P||q)+\text{KL}(q||P)>\text{KL}(q||P)+\text{KL}(q||P)$$
$$\frac{2\text{KL}(q||P)}{\text{KL}(P||q)+\text{KL}(q||P)}<1$$

Thus if $\sigma_p^2>\sigma_q^2$ then $\frac{2\,\text{KL}(q||P)}{\text{KL}(q||P)+\text{KL}(P||q)}<1$. Hence the theorem is proved.

**Corollary:** From Theorem 2 and 3: $\widetilde{\mathcal{F}}_{JSG}>\mathcal{F}_{KL}$ if $\sigma_p^2>\sigma_q^2$ and $\forall\,\alpha\in(0,1]$ such that $\alpha>\frac{2\,\text{KL}(q||P)}{\text{KL}(q||P)+\text{KL}(P||q)}$. Where, $P$ and $q$ are Gaussians and $P\neq q$.

Fig. 8 shows the sign of $\widetilde{\mathcal{F}}_{JSG}-\mathcal{F}_{KL}$ for various values of $\alpha$ and $\sigma_p-\sigma_q$. Two Gaussian distributions $P=\mathcal{N}(0,1)$ and $q=\mathcal{N}(0,\sigma_q^2)$ are considered for this purpose. It is evident that at least one value of $\alpha$ exists such that $\widetilde{\mathcal{F}}_{JSG}-\mathcal{F}_{KL}>0$ when $\sigma_p>\sigma_q$. In addition, for $\alpha=0$ or $\sigma_p=\sigma_q$, $\widetilde{\mathcal{F}}_{JSG}=\mathcal{F}_{KL}$.

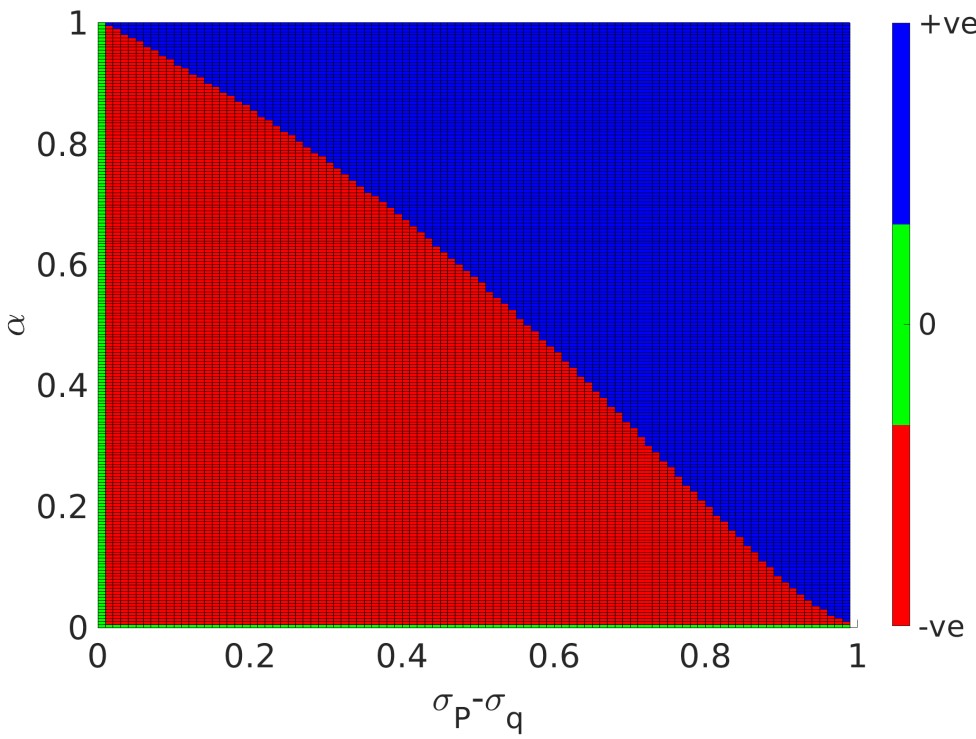

Figure 8: The sign of $\widetilde{\mathcal{F}}_{JSG} - \mathcal{F}_{KL}$ is plotted for various values of $\alpha$ and $\sigma_p - \sigma_q$.

## H   DATA SETS

### H.1   CIFAR-10

The Cifar-10 data set Krizhevsky et al. (2009) consists of 60,000 images of size $32 \times 32 \times 3$ belonging to 10 mutually exclusive classes. This data set is unbiased, with each of the 10 classes having 6,000 images.
Images were resized to $3 \times 224 \times 224$ pixels and normalized using the min-max normalization technique. The training data set was split into 80% 20% for training and validation respectively.

### H.2   HISTOPATHOLOGY

The histopathology data set Janowczyk & Madabhushi (2016); Cruz-Roa et al. (2014) is publicly available under a CC0 license at Paul Mooney (2017). The data set consists of images containing regions of Invasive Ductal Carcinoma. The original data set consisted of 162 whole-mount slide images of Breast Cancer specimens scanned at 40x. From the original whole slide images, 277,524 patches of size $50 \times 50 \times 3$ pixels were extracted (198,738 negatives and 78,786 positives), labeled by pathologists, and provided as a data set for classification.

The data set consists of a positive (1) and a negative (0) class. 20% of the entire data set was used as the testing set for our study. The remaining 80% of the entire data was further split into a training set and a validation set (80%-20% split) to perform hyperparameter optimization. The images were shuffled and converted from uint8 to float format for normalizing. As a post-processing step, we computed the complement of all the images (training and testing) and then used them as inputs to the neural network. The images were resized to $3 \times 224 \times 224$ pixel-wise normalization and complement were carried out as $p_n = (255 - p)/255$. $p$ is the original pixel value and $p_n$ is the pixel value after normalization and complement.

# I   RESULTS OF HYPERPARAMETER OPTIMIZATION

Results of the hyperparameter optimization for the two data sets are presented in Tables. 2 and 3. In Table. 2 Div denotes the divergence measure and LR is the learning rate. $\lambda$ is taken as 1 for the JS-G divergence-based loss function throughout this work.

Table 2: Cifar10 data set

| Div | Parameter | Noise level ($\sigma$) | | | | |
|---|---|---|---|---|---|---|
| | | **0.1** | **0.2** | **0.3** | **0.4** | **0.5** |
| KL | LR | 1e−4 | 1e−4 | 1e−4 | 1e−3 | 1e−3 |
| JS-G | $\alpha$ | 0.004 | 0.1313 | 0.2855 | 0.3052 | 0.2637 |
| | LR | 1e−4 | 1e−4 | 1e−4 | 1e−4 | 1e−5 |
| JS-A | $\lambda$ | 1000 | 1000 | 1000 | 1000 | 1000 |
| | $\alpha$ | 0.7584 | 0.6324 | 0.1381 | 0.6286 | 0.1588 |
| | LR | 1e−4 | 1e−4 | 1e−4 | 1e−3 | 1e−4 |

| Div | Parameter | Noise level ($\sigma$) | | | |
|---|---|---|---|---|---|
| | | **0.6** | **0.7** | **0.8** | **0.9** |
| KL | LR | 1e−3 | 1e−3 | 1e−3 | 1e−3 |
| JS-G | $\alpha$ | 0.2249 | 0.3704 | 0.3893 | 0.7584 |
| | LR | 1e−4 | 1e−4 | 1e−5 | 1e−3 |
| JS-A | $\lambda$ | 100 | 1000 | 1e4 | 1e5 |
| | $\alpha$ | 0.4630 | 0.1220 | 0.2282 | 0.5792 |
| | LR | 1e−3 | 1e−4 | 1e−5 | 1e−5 |

Table 3: Histopathology data set

| Divergence | $\alpha$ | $\lambda$ | Learning rate |
|---|---|---|---|
| KL | - | - | 1e−4 |
| JS-G | 0.0838 | 1 | 1e−4 |
| JS-A | 0.0729 | 100 | 1e−4 |

# J   ADDITIONAL EXPERIMENTS

We performed additional experiments to evaluate the performance of the proposed loss functions against various baselines. The results of these experiments are presented here.

## J.1   DETERMINISTIC NETWORKS

Bayesian CNNs with the proposed loss functions are compared with a deterministic CNN of the same architecture and the results are presented in Fig. 9. The deterministic CNN significantly overfits the training data and hence generalizes poorly for the validation dataset. Whereas, the Bayesian CNNs with their stochastic parameters can regularize much better than their deterministic counterparts. The validation accuracy of the deterministic CNN is about 8% lower than the proposed BNNs and 5% lower than the KL-based BNN. The difference between the training and validation accuracies are 63% and 86% less in KL- and JS-based BNNs respectively compared to the DNN, affirming greater generalization by KL- and JS-based BNNs.

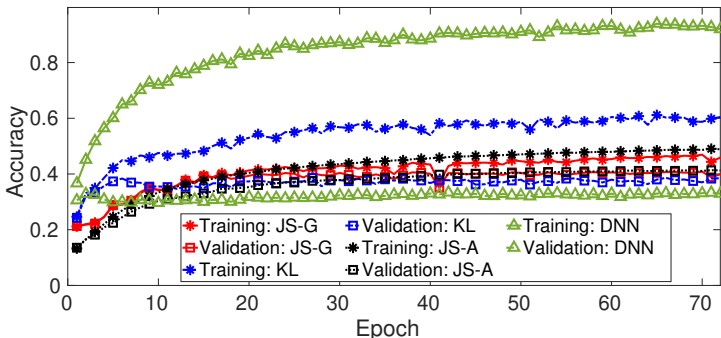

Figure 9: Performance of a deterministic CNN on Cifar-10 dataset with added Gaussian noise $\mathcal{N}(\mu = 0, \sigma = 0.9)$ .

## J.2 UNMODIFIED JS DIVERGENCES

The unmodified JS divergences in Eq. 1 and Eq. 2 fail to capture the dissimilarity between two distributions in the limiting cases of $\alpha$ as explained in the main text. However, we implemented these unmodified JS divergences in the loss function in Eq. 16 and compared their performance with the modified versions for completeness. From Fig. 10 it is evident that the modified JS divergences-based losses outperform the unmodified JS divergences-based losses on the histopathology dataset. The validation accuracy is less by about 3% and 2% for the unmodified JS divergences in Eq. 1 and Eq. 2 as compared to the modified ones respectively.

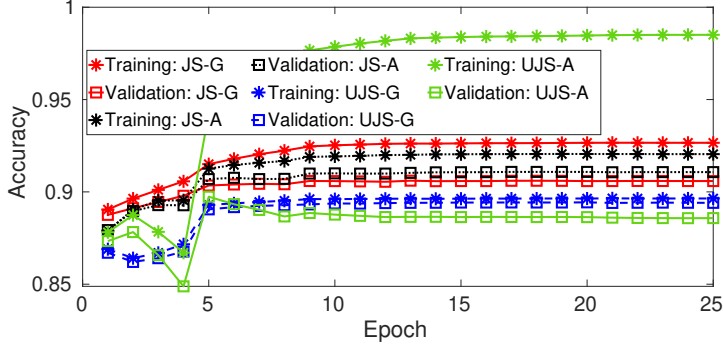

Figure 10: Performance of unmodified divergences (UJS) on histopathology

## J.3 $\lambda$ KL BASELINE

Implementing KL divergence in the constrained optimization framework (Eq. 16) yields a loss function which is henceforth called the $\lambda$KL baseline. We performed hyperparameter optimization on this baseline to tune the hyperparameters. The results of the $\lambda$KL baseline with the best-performing hyperparameters are presented in Fig. 11. We found that the maximum validation accuracy is 40.12% for this baseline on the Cifar-10 images with sigma=0.9. This validation accuracy is higher than the ELBO loss, which is 39.72 %, which corresponds to lambda=1. However, it is still less than the proposed JS loss, which has a maximum validation accuracy of 41.78%. In addition, for the lambda KL baseline, the regularization performance improves with lambda=68.9 although there are broad regions (between 20 to 90 epochs) where the network overfits.

The improved performance of JS divergences is a result of optimal penalization when the posterior is away from the prior. The functional form of the regularization term is now changed since the JS divergence is a weighted average of forward and reverse KL divergences. Through the alpha term of JS divergence, we can choose the optimal weights between the forward and reverse KL which is not present in the standard KL divergence. This combination of forward and reverse KL divergence

allows us to alter the shape of the multi-dimensional regularization term by adapting to the data, which is not possible to achieve by scalar multiplication of the regularization term.

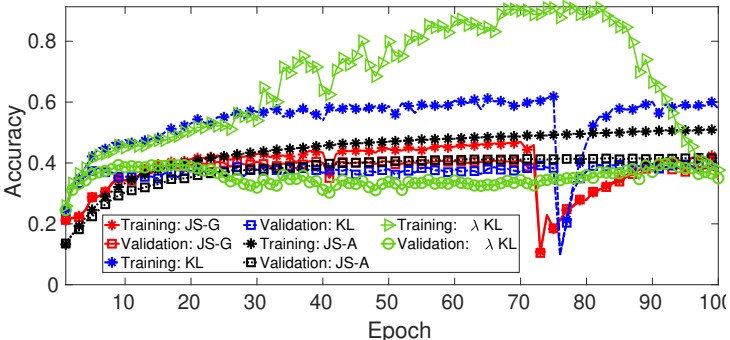

Figure 11: Performance of the $\lambda$ KL baseline for $\lambda = 68.9$.

### J.4 CIFAR-100 DATASET

The Cifar-100 data set Krizhevsky et al. (2009) consists of 60,000 images of size $32 \times 32 \times 3$ belonging to 100 mutually exclusive classes. Experiments on this dataset were carried out to evaluate the performance of the loss functions. The images were normalized using the min-max normalization technique and the training data set was split into 80% 20% for training and validation respectively. For this dataset, a ResNet18-V1 type architecture without the batch norm layers was used where the first two layers ( convolution and max pooling layer) are replaced with a single convolution layer with 3×3 kernel and 1×1 stride.

The results of the experiment are presented in Fig. 12. The test accuracies of KL, JS-G, and JS-A divergence-based losses were 22.81 %, 24.51 %, and 24.02 % respectively. Both the proposed JS divergence-based losses perform better than the KL loss in terms of test and validation accuracies. The regularization performance of the JS-A divergence was better than KL for this noise level. However, in terms of regularization the KL divergence performs better than the JS-G divergence for this dataset at the given noise level.

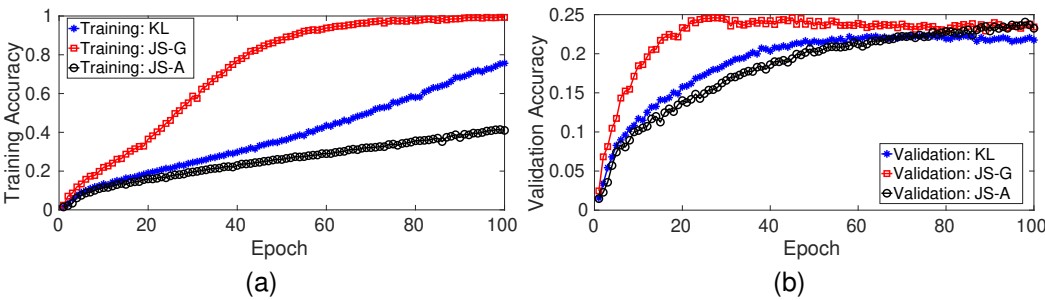

Figure 12: Performance comparison of the KL, JS-G, and JS-A divergence-based loss functions with Cifar-100 dataset with added Gaussian noise $\mathcal{N}(\mu = 0, \sigma = 0.5)$.

### J.5 UNCERTAINTY QUANTIFICATION

We performed UQ on the histopathology images by the standard BNN and proposed JS divergence-based BNNs. The means of the total uncertainties quantified by the JS-G and JS-A divergence-based BNNs are 74% and 29% greater than the KL-based BNN. However, these results need to be further analyzed to study the difference between the two.

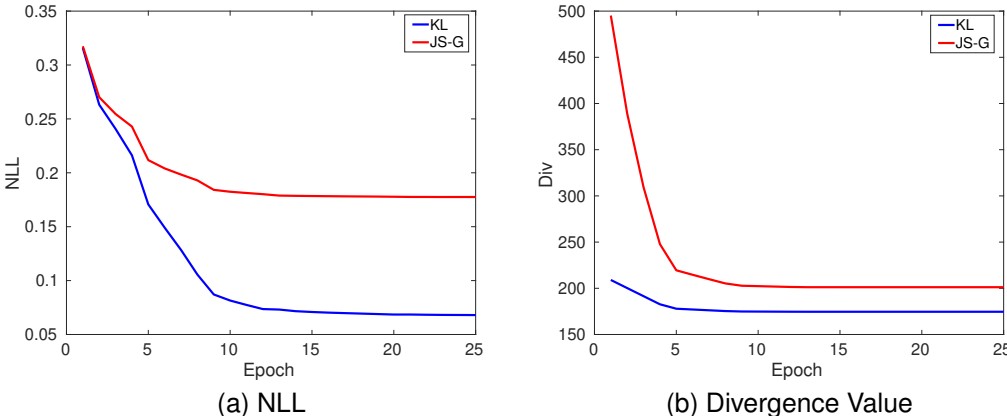

Figure 13: Comparison of the evolution of training loss for the histopathology dataset

## K    COMPUTATIONAL COST

All the convolutional neural networks presented in the paper were built on Python 3 using Apache MXNet with CUDA 10.2. Training, validation, and testing of all the networks were performed using the Nvidia Tesla V100 32GB GPUs.

A comparison of the computational time per epoch during training of the histopathology dataset is provided for the three loss functions in Table. 4. The number of MC samples for the KL divergence and JS-A divergence-based loss functions was taken as 1. Thus, the computational effort is almost equal for the MC sampled KL divergence and the closed-form evaluated JS-G divergence. Whereas, to evaluate the JS-A divergence both the prior and the posterior distributions of the parameters need to be sampled. Due to this, there is an increased computational effort to evaluate the JS-A divergence-based loss function which is reflected in the increased computational time in Table. 4

Table 4: A Comparison of computational time for the three loss functions.

| Divergence | Training Time per epoch (s) |
|---|---|
| KL | 1140 |
| JS-G | 1168 |
| JS-A | 1856 |

## L    EVOLUTION OF THE LOSS FUNCTIONS

The evolution of the divergence and negative log-likelihood part of the loss function during optimization is shown in Fig. 13 for the histopathology dataset. It is to be noted that the values are normalized by the number of test samples. The results are aligned with the theoretical results that the JS-G divergence penalizes higher than the KL divergence when the distribution q is farther from p. It is also seen that the negative log-likelihood is higher for the JS-G divergence as compared to the KL divergence in training. However, for the test dataset, the negative log-likelihood is lower for the JS divergences than the KL divergence which is desirable.

## M    OTHER PERFORMANCE EVALUATION METRIC

In this section, we compare the Test NLL, Test loss, and expected calibration error for the three losses for the histopathology dataset. From Table. 5 It is observed that the proposed JS divergences perform

better in terms of both ECE and negative log-likelihood than the KL divergence for the histopathology dataset. Note that the test loss and test NLL are normalized by the number of test samples. The JS-A loss is significantly higher because of the value of $\lambda = 100$.

Table 5: Performance comparison for the three loss functions.

| Divergence | Test NLL | Test loss | ECE |
|---|---|---|---|
| KL | 0.0810 | 177.6 | 0.0323 |
| JS-G | 0.0706 | 201.2 | 0.0158 |
| JS-A | 0.0689 | 15151.3 | 0.0091 |

## N    EXPERIMENTS ON REGRESSION

Regression experiments on multiple datasets are conducted following the framework of (Wan et al., 2020; Li & Turner, 2016). The network architecture is a single layer with 50 hidden units and ReLU activation function. We assume the priors to be Gaussians $p \sim \mathcal{N}(0, I)$ and 100 MC approximations are used to calculate the NLL part of the loss function for the KL and JS-G divergence-based losses, and 10 MC samples are used to evaluate the JS-A divergence based loss. The likelihood function is Gaussian with noise parameter $\sigma$ which is also learned along with the parameters of the network. Six datasets (Airfoil, Aquatic, Building, Concrete, Real Estate, and Wine) are evaluated and they are split into 90% and 10% for training and testing respectively. 20 trails are performed for each of these datasets and the average test root mean squared error and average negative log-likelihood for the JS divergences are compared with those models tested in (Wan et al., 2020). The results are provided in Table 6 and 7.

Table 6: Average root mean squared error. Except JS-G and JS-A all other values are taken from (Wan et al., 2020)

| Dataset | JS-G | JS-A | KL-VI | $\chi$-VI |
|---|---|---|---|---|
| Airfoil | 2.22±.25 | 2.32±.19 | **2.16±.07** | 2.36±.14 |
| Aquatic | 1.63±.19 | 1.13±.13 | **1.12±.06** | 1.20±.06 |
| Boston | 3.34±.88 | 2.91±0.73 | **2.76±.36** | 2.99±.37 |
| Concrete | 5.10±.67 | 4.88±.63 | 5.40±.24 | **3.32±.34** |
| Real Estate | **6.77±1.08** | 7.37±2.33 | 7.48±1.41 | 7.51±1.44 |
| Yacht | 0.82±0.35 | **0.69±.25** | 0.78±.12 | 1.18±.18 |

| Dataset | $\alpha$-Vi | TV-VI | $f_{c1}$-VI | $f_{c2}$-VI |
|---|---|---|---|---|
| Airfoil | 2.30±.08 | 2.47±.15 | 2.34±.09 | 2.16±.09 |
| Aquatic | 1.14±.07 | 1.23±.10 | 1.14±.06 | 1.14±.06 |
| Boston | 2.86±.36 | 2.96±.36 | 2.87±.36 | 2.89±.38 |
| Concrete | 5.32±.27 | 5.27±.24 | 5.26±.21 | 5.32±.24 |
| Real Estate | 7.46±1.42 | 8.02±1.58 | 7.52±1.40 | 7.99±1.55 |
| Yacht | 0.99±.12 | 1.03±.14 | 1.00±.18 | 0.82±.16 |

Table 7: Average negative log-likelihood. Except JS-G and JS-A all other values are taken from (Wan et al., 2020)

| Dataset | JS-G | JS-A | KL-VI | $\chi$-VI |
|---|---|---|---|---|
| Airfoil | 2.22±.09 | 2.72±.10 | **2.17±.03** | 2.27±.03 |
| Aquatic | 1.94±.13 | 1.78±.23 | **1.54±.04** | 1.60±.08 |
| Boston | 2.69±.31 | 3.3±1.12 | 2.49±.08 | 2.54±.18 |
| Concrete | 3.1±.17 | 3.11±.20 | 3.10±.04 | **2.61±.18** |
| Real Estate | **3.49±.06** | 3.56±.20 | 3.60±.30 | 3.70±.45 |
| Yacht | 1.51±.15 | **1.43±0.09** | 1.70±.02 | 1.79±.03 |
| **Dataset** | $\alpha$-Vi | TV-VI | $f_{c1}$-VI | $f_{c2}$-VI |
| Airfoil | 2.26±.02 | 2.28±.04 | 2.29±.02 | 2.18±.03 |
| Aquatic | 1.54±.07 | 1.56±.07 | 1.54±.06 | 1.55±.04 |
| Boston | **2.48±.13** | 2.51±.18 | 2.49±.13 | 2.51±.10 |
| Concrete | 3.09±.04 | 3.10±.05 | 3.09±.03 | 3.10±.04 |
| Real Estate | 3.59±.32 | 3.86±.52 | 3.62±.33 | 3.74±.37 |
| Yacht | 1.82±.01 | 1.78±.02 | 2.05±.01 | 1.86±.02 |

