# OpenReview forum: "Jensen-Shannon Divergence Based Novel Loss Functions for Bayesian Neural Networks"
_ICLR.cc/2024/Conference — Submitted to ICLR 2024_

### Official Review · Reviewer_uNB9 · 2023-10-18

**Soundness:** 2 fair
**Presentation:** 2 fair
**Contribution:** 2 fair
**Rating:** 3
**Confidence:** 4

**Summary:**

Variational inference (VI) has some issues when applied to Bayesian neural networks (BNNs). For instance, the optimization procedure of VI-BNNs is much harder than standard point estimation. This paper argues that this is because of the limitations of the KL divergence, in particular, due to its non-symmetric and unboundedness properties.

To alleviate this issue, the authors propose to use the JS divergence instead, which is bounded, symmetric, and generalizes the KL divergence. They view the ELBO objective as a constrained optimization objective and replace the KL in the regularization term with the JS. Theoretical study shows that the proposed JS loss induces a greater penalization (than the KL-based loss) as the variational posterior deviates away from the prior under some circumstances.

Finally, empirical findings show that the JS-based loss yields better generalization performances in both CIFAR-10 and Histopathology datasets.

**Strengths:**

The authors provide a solid discussion about VI-BNNs and about the recent advance in replacing the KL with other divergences. I also appreciate an extensive discussion about the JS and its variants (JS-G).

**Weaknesses:**

This paper falls short in several key areas (in no particular order): (i) motivation, (ii) presentation, and (iii) experiments.

**Motivation**

The authors repeatedly mentioned that the problem with the KL-divergence in VI-BNNs is because of its asymmetry and unboundedness, which leads to instability in optimization. I completely agree that instability is one of the glaring issues of VI-BNNs and it needs to be solved. However, when I read the paper, I don't see the connection between them---why do asymmetry and unboundedness lead to instability. This might be obvious for the authors but I don't think it is for the readers. I would suggest the authors either theoretically or empirically show the connection between them. In that way, the proposed JS-based loss can be much better motivated.


**Presentation**

I think the paper in general lacks polish---there are many typos or inconsistent naming (CIFAR vs Cifar, etc). But the main issues are at least two-fold for me. First, The authors derive the JS-based loss from the constrained optimization perspective, i.e., simply replacing the `KL(variational_posterior, prior)` with `JS(variational_posterior, prior)`. I'm sure this is correct since the form of `expected_nll + KL(variational_posterior, prior)` is derived from `KL(variational_posterior, true_posterior)`, i.e. it uses the fact that the KL is used (the KL also appears naturally when deriving the ELBO via the marginal likelihood). Put another way, I'm not sure if the proposed objective is still a valid ELBO. It would be great if the authors could discuss this.

Second, it is unclear what is the significance of the theoretical results in Sec. 3.4.2. The authors should discuss this and relate those results to practical applications, e.g. using the corollary for analytically picking the hyperparameter $\alpha$. But then again, this doesn't seem to be the case since in practice it is chosen via hyperparameter tuning.


**Experiments**

The experiments need to be expanded if this paper wants to convince practitioners. Currently, only two datasets and a single network are studied. Moreover, there is only a single baseline being compared, the standard KL-based VI-BNN---this is not a particularly strong baseline. It would be much more convincing if the authors compare practically relevant, strong BNN baselines like Laplace [1] and cyclical SGMCMC [2]. In particular, the accuracy on CIFAR-10 is very low for VI (~35%), whereas Laplace attains similar performance to the MAP-estimated network (for ResNet-18, around 92%+).

Moreover, generalization performance is just one side of the coin when talking about BNNs. Uncertainty quantification is the other side. Thus, the authors need to perform extensive experiments on UQ to make the paper complete.

**References**
[1] Daxberger et al., NeurIPS 2021
[2] Zhang et al., ICLR 2020

**Questions:**

Please see above.

---

> ### Author Response · Authors · 2023-11-23
> **Response to reviewer's comments**
>
> We thank the reviewer for providing valuable feedback, and thoughtful comments aimed at improving the manuscript. The reviewer's acknowledgment of the manuscript's strengths is encouraging. Additionally, we recognize and appreciate the constructive criticisms put forth by the reviewer. Below are the point-to-point responses addressing the reviewer's comments.
>
> ## Response to weaknesses pointed out
> We have addressed the inconsistencies in the revised version of the manuscript.
> 1. Motivation: Boundedness affects the optimization as seen in the figure in Appendix A. It shows that the KL and the JS-G divergence terms can go to infinity but the JS-A divergence is always bounded. Since the KL term can go to infinity during optimization, the loss function encounters singularity, and as a consequence, the optimization stops. Our implementation shows that the optimization fails at the first epoch when the KL divergence is evaluated numerically through MC sampling even for Gaussian priors.
> On the other hand, non-symmetry does not affect the stability of optimization. We seek symmetry because a symmetric divergence resembles a metric.
> 2. Presentation: \
> a)  The proposed objective maximizes the log-likelihood of observing the data with a constraint that the distribution learned is similar to the assumed prior distributions of the parameters. If we measure the similarity between the learned distribution q and the prior p using KL divergence then we obtain the ELBO. For the JS divergences, the proposed objective is not an ELBO in general. However, we can get back the ELBO from the proposed JS divergence-based losses with alpha = 0 and lambda = 1. \
> b) Theoretical results: The theoretical result (in sec 3.4.2) guarantees that there is an alpha between the mentioned ratio and unity for which the JS-G divergence is greater than the KL divergence for Gaussian distributions. The analytical result considers only the regularization and does not take the accuracy into account. Thus, the analytical result does not help to identify the alpha corresponding to the highest accuracy. Therefore, we seek the value of alpha that provides the best accuracy through hyperparameter tuning within the range (0,1). However, if the highest accuracy is approximately obtained for multiple values of alpha through hyperopt then the analytical result guides us to choose the highest value of alpha to ensure better regularization.
> 3. Experiments: \
> a) To implement the reviewer's comments we performed a comparison study on multiple regression datasets for various state-of-the-art VI techniques for BNNs. The results are provided in Appendix N of the revised manuscript. We found that the JS divergence-based losses perform as good as the other techniques or in some cases better in terms of both accuracy and log-likelihood for these datasets. The model’s accuracy is low due to the presence of a significant degree of noise (sigma=0.9). Similar performances are obtained for datasets with noise in the literature [Hendrycks et al., ICLR 2019 ]. With a lesser degree of noise, the accuracy significantly increases.
> b) To implement the reviewer’s suggestion, we performed UQ on the histopathology images by the standard BNN and JS divergence-based BNNs. The mean uncertainty quantified by the JS-G and JS-A divergence-based loss is 74% and 29% higher as compared to that of KL. However, these results need to be further analyzed to study the difference between the two. These additional results are reported in Sec.J.5 of the revised manuscript. A detailed study on the UQ aspects of the loss functions is an interesting question that we plan to address in future work.

---

### Official Review · Reviewer_kqYv · 2023-10-27

**Soundness:** 3 good
**Presentation:** 3 good
**Contribution:** 2 fair
**Rating:** 5
**Confidence:** 4

**Summary:**

In this paper, the authors have identified shortcomings in the KL divergence-based Variational Inference (VI) and have introduced loss functions based on two variants of the Jensen-Shannon divergence, referred to as JS-G and JS-A. Through both theoretical analysis and empirical experiments, they have substantiated that these newly proposed loss functions represent a more generalized and enhanced approach compared to the previous KL divergence-based VI methods.

**Strengths:**

Clarity
- The paper is well-crafted, making it easily understandable for readers.
- The theorems and proofs presented are straightforward and have a clear, comprehensible structure.

Originality and Significance
- This paper proposed new loss functions based on two variants of the Jensen-Shannon divergence, referred to as JS-G and JS-A which overcome some issues of the KL divergence-based loss function.
- They empirically validate the proposed loss functions show more improved generalization performance compared to KL divergence-based loss function.

**Weaknesses:**

Experiment
- The experiments conducted are somewhat basic and may not completely confirm the efficacy of the proposed loss functions. It is advisable to include additional experiments involving larger datasets and more extensive models.
- The model's performance on CIFAR10 and CIFAR100 datasets seems suboptimal, potentially due to the absence of batch normalization. It would be worthwhile to explore the impact of incorporating batch normalization and also consider using the Filter Response Normalization (FRN) layer if issues persist.
- It is recommended to provide ablation results concerning the parameters $\lambda$ and $\alpha" in the experimental analysis.

References

[1] Singh, S. and Krishnan, S. Filter response normalization layer: Eliminating batch dependence in the training of deep neural networks. In Proceedings of the IEEE/CVF Conference on Computer Vision and Pattern Recognition, pp. 11237–11246, 2020.

**Questions:**

Refer to weakness paragraph.

---

> ### Author Response · Authors · 2023-11-23
> **Response to reviewer's comments**
>
> We thank the reviewer for their valuable time and feedback to enhance the manuscript. The reviewer's recognition of the manuscript's strengths is uplifting for us. We equally acknowledge and value the criticisms raised by the reviewer. Below are the point-to-point responses addressing the reviewer's comments.
> ## Responses to weakness:
> 1. Lack of experiments: To implement the reviewer's comments we performed a comparison study on multiple regression datasets for various state-of-the-art VI techniques for BNNs. The results are provided in Appendix N of the revised manuscript. We found that the JS divergence-based losses perform as good as the other techniques or in some cases better in terms of both accuracy and log-likelihood for these datasets.
> 2. Suboptimal performance: The model’s performance is suboptimal due to the presence of a significant degree of noise (0.1<=sigma<=0.9). Similar performances are obtained for datasets with noise in the literature [Hendrycks et al., ICLR 2019 ]. With lesser degree of noise, the accuracy significantly increases.
> 3. Ablation results: Removing the parameters lambda and alpha is equivalent to setting alpha =0 and lambda=1. With this setting, KL divergence-based loss is obtained for the proposed loss functions. Therefore a separate ablation study is not performed in this work.

---

### Official Review · Reviewer_BhnF · 2023-10-30

**Soundness:** 3 good
**Presentation:** 3 good
**Contribution:** 2 fair
**Rating:** 5
**Confidence:** 4

**Summary:**

This paper proposes variational inference using an extended JS divergence as a solution to the problems with KL-based variational inference for BNNs. It introduces JS-A, an extension of original JS divergence itself, to address issues like divergent problems that arise when using JS divergence directly. Utilizing techniques from constrained optimization in variational inference, the paper proposes a new VI approach using JS-A and applies it numerically to problems of BNNs.

**Strengths:**

This paper proposes JS-A, an adaptation of JS divergence specifically tailored for Variational Inference (VI), rather than using JS divergence directly. Then the authors tried to theoretically clarify the properties of JS-A, which adds to its strength.

Additionally, the paper adopts a constrained optimization formulation instead of directly minimizing JS divergence, making the computation more manageable. This aspect demonstrates a thoughtful approach to addressing the challenges associated with VI.

**Weaknesses:**

The major concern is that I could not understand how the proposed objective function Eq.(17) is related to the original objective function Eq.(10). The existing constrained optimization formulation of VI, Eq(14), has the explicit relation to the KL minimization of Eq.(4).
However, I wonder whether any relation holds between Eq.(17) and (10). In the current version of the paper, two formulations seem irrelevant, i.e., Eq. (17) is the prior and posterior, and Eq(10) is the minimization between the true posterior and approximate posterior. Please elaborate on these relationships.

The next concern is the lack of the comparison between JS-A and JS-G. Although the theoretical properties are summarized in Table 1, I could not understand which is better when used in VI. From the theoretical properties, JS-A seems better than JS-G. However, when looking at numerical experiments, such as Fig. 12, JS-G seems better, especially for large-scale experiments. Could you add a comparison for JS-A and JS-G and discuss their differences as VI ?

**Questions:**

In Fig. 1, although the results of both JS-A and JS-G are presented, they are never discussed. What difference between JS-A and JS-G can we interpret from the figure ? Also, the author discussed only the mean. Please discuss the results of the variance in Fig. 1

Also, in Fig 1 (b) and (d), what is the difference between the red and black curves ?

In table 4, the proposed JS-A seems slower than the other methods in terms of wall clock time. Why ?

According to the explanation of Eq(18), the proposed method has both mode- and mean-seeking properties. I think this resembles alpha divergence VI. Could you show the comparison of alpha-divergence VI numerically ?

---

> ### Author Response · Authors · 2023-11-23
> **Response to reviewer comments**
>
> We thank the reviewer for their time and insightful suggestions to enhance the manuscript. The reviewer's recognition of the manuscript's strengths is encouraging, and we also value their constructive criticisms. Below, we present our point-to-point responses addressing the reviewer's comments.
>
> ## Responses to the weakness pointed out
> 1. Since Eq 10 cannot provide a tractable loss function, we use the constrained optimization approach to derive the loss. Eq 10 and 17 are not directly related and neither can be derived from another. We present Eq(10) to show the fact that the minimization between true and variational posterior using JS divergence is intractable. Expanding Eq. (10) we obtain the following equation, which is henceforth referred to as Eq. (10)
> $$F_{JSG} = (1-\alpha)^2 E_{q(\mathbf{w}|\boldsymbol{\theta})} \left[ \log \frac{q(\mathbf{w}|\boldsymbol{\theta})}{P(\mathbf{w})} \right] - E_{q(\mathbf{w}|\boldsymbol{\theta})}\left[\log P(\mathbb{D}|\mathbf{w})\right] + (2 \alpha - \alpha^2)E_{q(\mathbf{w}|\boldsymbol{\theta})}\left[\log P(\mathbb{D}|\mathbf{w})\right] + \alpha^2 E_{P(\textbf{w}|\mathbb{D})} \left[  \log \frac{P(\textbf{w}|\mathbb{D})}{q(\textbf{w}|\boldsymbol{\theta})}\right] $$
> The above equation can be compared with Eq. (17)
> $$\widetilde F_{JSG} = (1-\alpha)^2 E_{q(\mathbf{w}|\boldsymbol{\theta})} \left[ \log \frac{q(\mathbf{w}|\boldsymbol{\theta})}{P(\mathbf{w})} \right]  - E_{q(\mathbf{w}|\boldsymbol{\theta})}\left[\log P(\mathbb{D}|\mathbf{w})\right] +  \alpha^2 E_{P(\mathbf{w})} \left[ \log \frac{P(\mathbf{w})}{q(\mathbf{w}|\boldsymbol{\theta})} \right] $$
> We see that the first two terms of Eq. (10) and Eq. (17) are identical. The fourth term of Eq. (10) involves an expectation over the unknown posterior probability P(w|D) which makes it intractable. Whereas, the constrained optimization framework results in Eq. (17) that involves an expectation over the prior probability P(w), which is tractable.
> We can not obtain the last term of Eq. (17) from the last two terms of Eq. (10) by applying Bayes’ theorem as shown below.
> $$ (2 \alpha - \alpha^2) E_{q(\mathbf{w}|\boldsymbol{\theta})}\left[\log P(\mathbb{D}|\mathbf{w})\right] + \alpha^2 E_{P(\mathbf{w})} \left[ \frac{P(\mathbb{D}|\textbf{w})}{P(\mathbb{D})} \log \frac{P(\mathbb{D}|\textbf{w}) P (\textbf{w})}{q(\textbf{w}|\boldsymbol{\theta})P(\mathbb{D})}\right] \neq \alpha^2 E_{P(\mathbf{w})} \left[ \log \frac{P(\mathbf{w})}{q(\mathbf{w}|\boldsymbol{\theta})} \right] $$
> 2. Both these divergences have advantages and limitations. The major difference is that JS-A is bounded but doesn't provide a closed-form expression, unlike JS-G which is unbounded but has closed form expression when p and q are Gaussians. Because of the presence of analytical expression, the JS-G-based loss is faster to train as seen in the time comparison results. On the other hand, JS-A eliminates the instability in optimization due to its boundedness property. Both these loss functions can generalize better as shown in the results.
>
> ## Response to the questions
> 1. Fig 1 shows the higher penalization of the JS divergence-based loss functions when the posterior moves away from the prior distribution. The value of JS-G>KL when sigma_p > sigma_q. The variance figure of JS-G is a depiction of theorem 3. It shows the change of JS-G with respect to KL as sigma is changed.  In the case of JS-A divergence, the value of KL>JS-A when lambda = 1. However, by increasing lambda, we obtain a higher penalization of the loss which is depicted in Fig 1. Sec.3.4.1 and Sec 3.4.2 of the revised manuscript are modified to reflect these comments.
> 2. Red curves are with lambda =1 and black curves with lambda>1. For the value of lambda = 1, the value of the JS-A divergence is smaller than the KL divergence denoted by red curves. With lambda>1 the values of the JS-A are scaled as shown in red curves.
> 3. The JS-A divergence is slower because it does not have a closed-form expression and it has to be implemented through MC sampling as mentioned in Appendix K of the manuscript.
> 4. To implement the reviewer's comments we performed a comparison study on multiple regression datasets for various state-of-the-art VI techniques for BNNs including alpha divergence VI. The results are provided in Appendix N of the revised manuscript. We found that the JS divergence-based losses perform as good as the other techniques or in some cases better.

---

### Official Review · Reviewer_AJUX · 2023-10-31

**Soundness:** 1 poor
**Presentation:** 2 fair
**Contribution:** 2 fair
**Rating:** 3
**Confidence:** 4

**Summary:**

- This paper proposed a new objective function for variational inference (VI) in Bayesian Neural Networks (BNN). This objective function is built upon a modified version of the generalized Jensen-Shannon (JS) divergence, denoted as the JS-A divergence. Unlike the Kullback-Leibler (KL) divergence, the JS-A divergence has an upper bound and exhibits symmetry when the weight of the arithmetic mean, denoted as $\alpha$, is set to $0.5$.
- The authors theoretically confirmed the effectiveness of their proposed objective by comparing the regularization terms of their objective with those of the KL-based VI objective.
- The authors validated the predictive accuracy of their method using various benchmark datasets.

**Strengths:**

- This paper tries to expand the choices for objective functions in VI for BNN. It places particular emphasis on exploring the JS divergence family, which has several intriguing properties.
- The authors confirmed the performance of their proposed method through experiments carried out on BNN with highly sophisticated model architectures, including Convolutional Neural Networks (CNN) and ResNet-18.

**Weaknesses:**

I would like to express my sincere respect for all the efforts the authors have invested in this paper.
Unfortunately, however, I cannot strongly recommend this paper as an ICLR 2024 accepted paper for the following reasons: (1) concerns regarding the validity of the rationale for the problem statement, (2) the validity of changing the divergence to ensure boundedness, and (3) the lack of sufficient experimental data to justify the contributions made.
The details are provided below. If there are any misunderstandings, I apologize, and I would appreciate it if you could explain them to me.

## Concerns regarding the validity of the rationale for the problem statement
- In this paper, the authors adopted the problem that the KL-based VI objective used in BNN, such as [Blundell et al., 2015], is unbounded and lacks symmetry, which negatively impacts optimization stability and generalization performance. As the basis for this assertion, they reference [Hensman et al. (2014); Dieng et al. (2017); Deasy et al. (2020)]. However, in my opinion, these references do not explicitly suggest that the unbounded and non-symmetric nature of the objective affects generalization performance. For instance, [Dieng et al. (2017)] identify the decrease in generalization performance due to **the underfitting of posterior variance** by KL-based VI and demonstrate that minimizing the upper bound based on the chi-squared distribution (called CUBO) can resolve this issue, and it is not stated that this underfit of posterior variance is caused by the unbounded and asymmetric properties of KL objective. This raises significant concerns regarding the validity of the problem setting adopted in this paper unless it is theoretically and/or empirically demonstrated that the unbounded and non-symmetric objectives affect the stability of optimization and generalization performance of VI. There still be room for reconsidering whether the issues of divergence boundedness and asymmetry should be approached in the context of BNN+VI rather than in the context of divergence measures for distributions.
- The authors mentioned that they have proposed two new objective functions under the problem setting described above. However, according to Table 1, one of them (JS-G) does not eliminate boundedness. This point is not addressed in either the contributions section or the limitations section.

## The validity of changing the divergence to ensure boundedness
- If we utilize several practical approaches proposed in the context of PAC-Bayes, such as those suggested by [DR18] and [P21] (bounded cross entropy) or appropriately initializing model weights and prior, it becomes possible to bound the KL-based objective in the case of the Gaussian distribution that is the focus of this paper. Therefore, it is necessary to carefully consider whether the approach to resolving the issue of unbounded KL-based objectives by altering the upper-bounded divergence is valid. However, the merits of the proposed method using JS divergence have not been discussed in comparison to these approaches, thus the validity of the approach remains unconfirmed.

## The lack of sufficient experimental data to justify the contributions made.
- If my understanding is correct, Theorems 2 and the corollary demonstrate that the proposed objective function becomes tight. However, as numerical experiments to verify this have not been conducted, the validity of the theoretical analysis remains unconfirmed. In this regard, plotting the progression of ELBO and the proposed objective values for each iteration would quickly provide confirmation.
- A comparison regarding the regularization capacity is briefly presented in Figure 1; however, this verification has not been conducted in experiments on the other benchmark datasets. To justify the contributions of this paper based on the relationship between regularization capacity and generalization performance, it is necessary to confirm, for example, the transition of KL and JS divergences during the learning process and the correlation between the final obtained divergence values and predictive performance. Currently, there are not enough experimental results provided to fully support the effectiveness of the proposed method.
- The authors solely compare the performance of their proposed method and existing methods based on predictive accuracy. However, since BNNs are probabilistic models, it is crucial to consider how well they capture the uncertainty in predictions, which is an important performance metric in many BNN studies, as confirmed experimentally by researchers such as [I21, P21]. To claim state-of-the-art performance, a more comprehensive performance evaluation on the test data, including metrics like test-ELBO, test-NLL, and Expected Calibration Error (ECE), is necessary.
- There is no information about how predictions are being made. Is this an evaluation based on deterministic predictions using the mean parameters of the variational distribution? Or is it a probabilistic prediction through Bayesian model averaging (BMA) using samples from the variational distribution? Since BNNs are probabilistic models, performance evaluation through BMA is essentially necessary.
- I have some concerns regarding the hyperparameter setting, particularly $\lambda$, in the existing KL-based VI objective for BNNs ([Blundell et al., 2015]). This aspect is also related to what is referred to as the *posterior temperature* [A21]. In many contexts, such as [Blundell et al., 2015] and [P21], $\lambda$ is commonly set as $\lambda = \frac{1}{n}$ using the training data size $n$ to balance the objective. However, in this paper, $\lambda$ is set to $\lambda=1$. This setting can potentially lead to an explosively large KL divergence in BNNs, which may adversely impact the final performance. Consequently, this experimental setup is considered unfavorable when compared to existing methods, raising doubts about the validity of the conclusion that the proposed method outperforms others.
- The initial values for network parameters are taken from pre-trained neural networks on ImageNet. However, in actual experiments, the authors use datasets such as CIFAR-10/100 and they do not employ ImageNet. This might seem somewhat unusual, as the reasons for this decision are not clearly articulated.

## MISC
- Due to factors like typos, the readability has been compromised. Please make an effort to refine the overall presentation. Here are a few examples of corrections.
- Sec.1, 2th paragraph: What does it mean "the two most commonly used techniques to approximate them (i.e., Bayesian inference) are the Variational Inference (VI)?"
- Sec.1, 2th paragraph: Citation for VI and MCMC could be required, e.g., [JGJS99]
- Sec.3.1, the first sentence: The Generalized JS... --> The generalized JS...
- Sec.4.1, the first sentence: ...on two Data sets... --> ... on two data sets...

## Citation
(Note: I am not the author of the following papers)

[DR18]: Gintare Karolina Dziugaite and Daniel M. Roy. Data-dependent PAC-Bayes priors via differential privacy. In Advances in Neural Information Processing Systems, pages 8440–8450, 2018.

[P21]: María Pérez-Ortiz, Omar Rivasplata, John Shawe-Taylor, and Csaba Szepesvári. Tighter Risk Certificates for Neural Networks. Journal of Machine Learning Research, 22(227):1−40, 2021.

[I21]: Alexander Immer, Matthias Bauer, Vincent Fortuin, Gunnar Rätsch, and Mohammad Emtiyaz Khan. Scalable Marginal Likelihood Estimation for Model Selection in Deep Learning. In ICML2021

[A21]: Laurence Aitchison. A statistical theory of cold posteriors in deep neural networks. In ICLR2021.

[JGJS99]: M. I. Jordan, Z. Ghahramani, T. S. Jaakkola, and L. K. Saul. An introduction to variational methods for graphical models. Machine Learning, 37(2):183–233, 1999.

**Questions:**

In connection with the weaknesses mentioned above, I would like to pose several questions related to the concerns raised.
I would appreciate your responses.

- Is there any research providing a detailed discussion on how the non-unboundedness and non-symmetry of KL-based methods can affect the stability and generalization performance of optimization? If not, can you provide an explanation or theoretical and empirical evidence to support this fact?
- In terms of boundedness, it's possible to keep KL divergence bounded without using JS divergence by employing clever techniques such as setting appropriate initial values for the weights of priors and models or using techniques like those found in [DR18; P21] to bound cross-entropy. What sets the proposed method apart in this regard?
- How does the objective value evolve during optimization? Does it align with the theoretical results, with that of the proposed method tending to be smaller?
- What are the results for ELBO, NLL, and ECE on the test dataset?
- In the experiments, are you evaluating deterministic predictions using the mean parameters of the variational distribution, or are you using BMA through sampling from the variational distribution? In the context of BNNs, performance evaluation with BMA is necessary.
- Why did you not adopt the setting $\lambda=\frac{1}{n}$ as typically handled in Bayes-by-Backprop [Blundell et al., 2016]?
- It seems that hyperparameter $\alpha$ is being optimized using Hyperopt. How sensitive is the model's performance to this setting? If sensitivity could indicate instability, it would be one of the limitations.
- Why did you choose to initialize the parameters of the BNN with a pre-trained neural network on ImageNet, even though experiments were not conducted on this dataset?


================ AFTER REBUTTAL & DISCUSSION ================

Due to the timing of receiving the rebuttal just before the deadline, I have not had adequate time to thoroughly review the revised manuscript. In my view, several issues, including those raised by both myself and other reviewers—such as the validity, motivation, and properties of employing loss functions based on bounded divergence, as well as appropriate experimental settings—still demand careful consideration and discussion.
Therefore, I keep my score.

**Details Of Ethics Concerns:**

I believe that this work does not raise any ethical concerns because it is a methodological study focused on VI for BNNs.

---

> ### Author Response · Authors · 2023-11-23
> **Response to reviewer comments**
>
> We express our gratitude to the reviewer for their detailed review, valuable feedback, and constructive suggestions to enhance the manuscript. We are pleased that the reviewer recognizes the strengths of our work, which encourages us to further improve it. Moreover, we value the reviewer's criticisms, and in response, we have provided detailed point-to-point responses to address their comments.
>
> ## Responses to the weaknesses pointed out
> 1. Concerns regarding the validity of the rationale:
>  Boundedness affects the optimization as seen in the figure in Appendix A. It shows that the KL and the JS-G divergence terms can go to infinity but the JS-A divergence is always bounded. Since the KL term can go to infinity during optimization, the loss function encounters singularity, and as a consequence, the optimization stops. Our implementation shows that the optimization fails at the first epoch when the KL divergence is evaluated numerically through MC sampling even for Gaussian priors. We have not observed any direct effect of lack of boundedness on the generalization performance.
> Non-symmetry does not affect the stability of optimization. We seek symmetry because a symmetric divergence resembles a metric. However, we agree with the reviewer that lack of symmetry does not affect the stability of optimization and generalization performance.
> The Introduction section of the manuscript is changed to reflect these facts.
> 2. Other ways to keep KL divergence bounded:
> Yes, it is possible to keep the KL divergence bounded by selecting appropriate priors. However, that imparts restrictions on the priors that can be used. Techniques found in [DR18; P21] provide bounds on the KL divergence only for specific distributions (exponential distributions) whereas, the boundedness of JS-A divergence shown in the present work is true for any arbitrary distributions. In addition, bounding the cross-entropy loss ( [DR18; P21]) does not resolve the unboundedness of the divergence part of the loss function, which is the goal of this work.
> 3. Evolution of objective:
> To implement the reviewer's suggestion, the evolution of the divergence and NLL part of the loss function during optimization is shown in Fig 13 in Appendix L of the revised manuscript for the histopathology dataset. It is aligned with the theoretical results that the JS-G divergence penalizes higher than the KL divergence when the distribution q is farther from p. It is also seen that the negative log-likelihood is higher for the JS-G divergence as compared to the KL divergence. However, for the test dataset, the negative log-likelihood is lower for the JS divergences than the KL divergence which is desirable.
> 4. Other performance metrics:
>  To implement the reviewer’s suggestion we now compare the Test NLL, Test loss, and ECE for the three losses for the histopathology dataset. This is provided in Table 5 in Appendix M. It is observed that the proposed JS divergences perform better in terms of both ECE and negative log-likelihood than the KL divergence for the histopathology dataset.
> 5. BMA:
> Samples from the posterior distribution are considered while training the Bayesian NN. However, we are predicting the labels of the test set using the mean of the parameters. To implement the reviewer’s suggestion we have compared the predictions obtained with the mean and the BMA. The result shows that the predictions by both approaches differ by about 0.006 % in accuracy for the histopathology dataset.
> 6. Hyperparameter setting:
>  Bayes-by-backprop uses a factor that depends on the mini-batch size (M) while minimizing ELBO in mini-batches. A factor of 1/M is considered for all three losses while mini-batches are used in this work as well. The mini-batch implementation was not provided in the previous version which is now added in Appendix D of the revised manuscript.
> 7. Sensitivity to hyperparameters:
> We did not observe any dramatic changes in the performance for small changes in parameters. Therefore, based on our empirical experiments the present model is not sensitive to these hyperparameters.
> 8. Initialization of the network:
> This was done to get a better initialization of the weights of the network for faster convergence. However, the network parameters are not frozen and hence are allowed to be learned during optimization. We found that initializations with pre-trained network weights performed better than random initialization.
>
> ## Misc
> All the miscellaneous comments are addressed in the revised manuscript.

---

> > ### Comment · Reviewer_AJUX · 2023-11-23
> > **Acknowledgement**
> >
> > Thank you for your thoughtful response to my concerns. Regrettably, given the imminent deadline, there isn't sufficient time to thoroughly assess all your responses and review the revised manuscript. I am of the opinion that several issues, including those raised by myself and other reviewers, such as the appropriateness of utilizing loss functions based on bounded divergence, motivation, properties, and suitable experimental settings, still need careful consideration and discussion.
> >
> > Anticipating that substantial revisions and additional analysis will be necessary, I believe the manuscript should undergo another round of peer review at the other conferences. Consequently, I intend to maintain my current review score.

---

### Official Review · Reviewer_PkhQ · 2023-11-02

**Soundness:** 2 fair
**Presentation:** 1 poor
**Contribution:** 2 fair
**Rating:** 3
**Confidence:** 4

**Summary:**

The authors propose novel VI learning methods using variants of JS divergence instead of the usual KL divergence. The authors demonstrate that the learning can be performed using gradient-based and MCMC-based optimization methods.
The proposed methods seems to provide a useful regularization when training a BNN for the classification tasks in the paper.
The authors demonstrate the advantage of the method for noisy and biased data.

**Strengths:**

* The authors derive a a regularization term and justify it using statistical considerations, which I like.
* The final loss is rather clean and intuitive.

**Weaknesses:**

* The loss relies on 2 hyper-parameters (lambda, alpha) which might make the practical use of the method problematic (e.g., in cases where hyper-parameters search is too expensive).
* It is unclear if simpler regularization methods (e.g., dropouts) might yield similar improvement.
* Experimentation is limited to two simple datasets.
* Comparison to other simpler regularization methods (e.g., dropout) is missing.

**Questions:**

Below are comments and questions.

Comments:
* In the introduction: The number of parameters has nothing to do with the ability to provide a robust measure of uncertainty. The model choice allows such ability.
* In the introduction: The variational distribution is learned by minimizing its dissimilarity with respect to the true posterior  - VI does not minimize this quantity since the true posterior is unknown. It can be derived starting from the KL divergence between the approximate and true posterior though.
* Derivation of Eq. 5 from Eq. 4 is awkward and the justification for dropping p(D) is not clear (being a constant that multiplies the approximate posterior). Typically derivation entails the use of Jensen’s inequality which also facilitate the relation between Eq. 5 and Eq. 4.

Questions:
* Near Eq. 2: Gα(x, y) = x1−αyα  - G_alpha is an unnormalized distribution in the general case, is that true?
* Near Eq.3: JS-G(p||q)|α = JS-G(p||q)|1−α  type, should it be non equality?
* Figure 2: it is unclear wether other regularization techniques (like dropouts) could narrow down the gap for VI with KL divergence, without the need for the forward KL regularization term in the loss. Also, 2b looks like overfitting for the VI with KL divergence. Did you try other regularization methods?

---

> ### Author Response · Authors · 2023-11-23
> **Response to reviewer's comment**
>
> We thank the reviewer for providing feedback, insightful comments, and valuable suggestions for improvements to the manuscript. We are encouraged by the reviewer’s appreciation of the strengths of the manuscript. We also appreciate the criticisms from the reviewer. Following are the point-to-point responses to the reviewer’s comments.
>
> ## Response to the weaknesses pointed out:
> 1. Additional hyperparameters:
> This fact is mentioned in the limitation section of the manuscript. When alpha = 0 and lambda =1, we get back the KL divergence-based loss. This can be utilized in such cases where hyper-parameters are expensive. However, our empirical experiments show that finding the optimal values of the two additional parameters is not significantly more expensive. The addition of these two new hyperparameters to the existing set of hyperparameters (such as learning rate and batch size) does not add significant computational cost.
> 2. Comparison to regularization methods:
> The key contribution of this work is to overcome the unboundedness due to KL divergence in the state-of-the-art BNNs. In doing this, we observe better regularization through theoretical and empirical analysis of the loss functions. Since the motivation for this work is not to improve the regularization performance, a comparison with other regularization methods is not performed.  In addition, BNNs automatically regularize through model averaging and the prior distribution. For example, when Gaussian prior is used, it yields L2 regularization.
> 3. Limited experiments:
> To implement the reviewer's comments we performed a comparison study on multiple regression datasets for various state-of-the-art VI techniques for BNNs. The results are provided in Appendix N of the revised manuscript. We found that the JS divergence-based losses perform as good as the other techniques or in some cases better in terms of both accuracy and log-likelihood for these datasets.
>
> ## Response to questions:
> ### Comments:
> All the comments by the reviewer are addressed in the revised manuscript.
> 1. Changed to: due to deterministic parameters, CNNs cannot provide a robust measure of uncertainty.
> 2. Changed to: The variational distribution is learned by minimizing an objective function derived from its dissimilarity with respect to the true posterior
> 3. Modified to explain it better
>
> ### Questions:
> 1. Yes, the geometric JS divergence is unnormalized in the general case.
> 2. This is equality. Symmetry in this context means JS-G(p||q)|α = JS-G(q||p)|α
> 3. BNNs automatically regularize through model averaging and the prior distribution as mentioned earlier. Therefore, we did not try additional regularization methods.

---

### Author Response · Authors · 2023-11-23

We thank all the reviewers for providing feedback, insightful comments, and valuable suggestions for improvements to the manuscript. We are encouraged by the reviewers' appreciation of the strengths of the manuscript. We also appreciate the criticisms from the reviewers. We have added the point-to-point responses to the reviewers' comments individually and revised the manuscript to reflect the suggestions.

---

### Meta-Review · Area_Chair_SraQ · 2023-12-06

**Metareview:**

This paper claims that using the KL divergence in VI for BNNs poses some problems and proposes to overcome them using a novel objective based on the Jensen-Shannon divergence. After an active discussion between authors and reviewers, all reviewers agreed to reject this paper. While the reviewers praised the importance of the problem and the elegance and thoughtfulness of the approach, they were critical of the presentation of the manuscript, the motivation for the approach, and the experiments. I believe that this could be a really interesting contribution and would encourage the authors to take the reviewer feedback seriously and resubmit a revised version of the paper in the future.

**Justification For Why Not Higher Score:**

the reviewers believe that the paper could be significantly improved for a resubmission

**Justification For Why Not Lower Score:**

N/A

---

### Decision · Program_Chairs · 2024-01-16

Reject